# Oxidative metabolisms catalyzed Earth's oxygenation

Haitao Shang [1,2 ✉], Daniel H. Rothman [1,2] & Gregory P. Fournier [2]

The burial of organic carbon, which prevents its remineralization via oxygen-consuming processes, is considered one of the causes of Earth's oxygenation. Yet, higher levels of oxygen are thought to inhibit burial. Here we propose a resolution of this conundrum, wherein Earth's initial oxygenation is favored by oxidative metabolisms generating partially oxidized organic matter (POOM), increasing burial via interaction with minerals in sediments. First, we introduce the POOM hypothesis via a mathematical argument. Second, we reconstruct the evolutionary history of one key enzyme family, flavin-dependent Baeyer–Villiger mono-oxygenases, that generates POOM, and show the temporal consistency of its diversification with the Proterozoic and Phanerozoic atmospheric oxygenation. Finally, we propose that the expansion of oxidative metabolisms instigated a positive feedback, which was amplified by the chemical changes to minerals on Earth's surface. Collectively, these results suggest that Earth's oxygenation is an autocatalytic transition induced by a combination of biological innovations and geological changes.

[1] Lorenz Center, Massachusetts Institute of Technology, Cambridge, MA 02139, USA. [2] Department of Earth, Atmospheric, and Planetary Sciences, Massachusetts Institute of Technology, Cambridge, MA 02139, USA. ✉email: htshang.research@gmail.com

The abundance of molecular oxygen ($O_2$) is a remarkable feature of Earth's atmosphere, but it remains unclear why and how Earth evolved from the ancient $O_2$-deficient environment to the modern $O_2$-rich world[1,2]. Geochemical studies suggest that $O_2$ concentrations significantly increased during the Great Oxidation Event (GOE), around 2400 to 2300 million years ago (Ma)[1,2]. $O_2$ is produced by oxygenic photosynthesis and consumed by aerobic respiration or oxidation of reducing compounds[1,2]. $O_2$ accumulates when its production rate exceeds its consumption rate[3]. For example, it has been suggested that the GOE might have resulted from the oxidation of Earth's mantle or crust, which reduced the $O_2$ consumption rate; the dramatic burial of organic matter, which increased the $O_2$ production rate; or other mechanisms[1–3].

Most models[4–6] envision the rise of $O_2$ as a shift in the equilibrium of the global redox state, essentially following Le Chatelier's principle at a planetary scale. However, an alternative possibility exists: the global redox state exhibits multiple equilibria, and oxygenation occurs when $O_2$ levels dynamically switch from one stable state to another. Such a shift between alternative stable states requires the existence of one or more positive feedbacks[7]. Although purely geochemical feedbacks are possible[8–10], they may be less responsive to environmental changes compared to geochemical feedbacks intertwined with biological evolution[11].

Here we suggest that the expansion of oxidative metabolisms provided a positive feedback for the GOE. This may appear counter-intuitive: oxidative metabolic processes, after all, consume $O_2$. A potentially important positive feedback nevertheless lies in the interaction of oxidized metabolic products with minerals in sedimentary environments. We suggest that, in low-$O_2$ environments, incomplete degradation by oxidative metabolisms results in partially oxidized organic matter (POOM) that is persistently protected by minerals. POOM may then be immobilized in sedimentary rocks, thereby enhancing the accumulation of atmospheric $O_2$. In modern sediments, POOM is produced by aerobic metabolisms catalyzed by oxidative enzymes[12]; here we focus on a representative, POOM-producing enzyme—flavin-dependent Baeyer–Villiger monooxygenase (BVMO)[13]. We reconstruct the evolutionary history of BVMOs to determine if its evolution coincides with Earth's oxygenation. Our analyses show that one major group of marine POOM producers (i.e., SAR202 bacteria) likely acquired BVMO around the time of the GOE via a horizontal gene transfer event, and that subsequent diversification of this lineage, as well as the BVMO gene family, track the major periods of oxygenation in the Proterozoic and Phanerozoic.

We begin with a brief analysis of mechanisms underlying the stability of modern $O_2$ levels and the need for a mechanism of destabilization near the time of the GOE. We then introduce the POOM hypothesis and quantitatively determine the conditions under which it applies. Phylogenetic methods are then applied to show the relevance of the oxidative metabolisms to the rise of atmospheric $O_2$. Finally, we discuss how the expansion of oxidative metabolisms and the evolution of Earth's surface environment together led atmospheric $O_2$ to shift to higher levels.

## Results

**Stability of atmospheric $O_2$ in the modern environment.** Organic matter is highly heterogeneous[14]; so too is the sedimentary environment in which it is deposited[15]. Compositional and environmental factors both help determine how quickly organic matter decays, and how much is ultimately preserved[15]. To better understand how mechanisms of preservation influence the stability of $O_2$ levels, we construct a model of heterogeneity that considers only two types of organic matter: a "labile"

component $g_1$ that ultimately always decays and is never preserved, and a "recalcitrant" component $g_2$ that decays only in the presence of $O_2$[16]. We assume that $g_1$ degrades with a rate constant $k_1$ while $g_2$ degrades (in oxic sediments) with a slower rate constant $k_2 < k_1$. Organic degradation rate is usually expressed in terms of first-order kinetics[15]. The aerobic degradation rates at time $t$ are then written as

$$\frac{dg_1}{dt} = -k_1 g_1, \qquad t \geq 0 \qquad (1)$$

$$\frac{dg_2}{dt} = -k_2 g_2, \qquad 0 \leq t \leq t_{ox}, \qquad (2)$$

where $t_{ox}$, the *oxygen-exposure time*[17], is the time over which organic matter is exposed to $O_2$. We denote the total amount of organic carbon initially deposited in sediments by $g_0 = g_1(0) + g_2(0)$ and the initial fraction of $g_2(0)$ by $a$. The initial conditions then read

$$g_1(0) = (1 - a)g_0 \quad \text{and} \quad g_2(0) = ag_0. \qquad (3)$$

When $t > t_{ox}$, the remaining $g_1(t_{ox})$ and $g_2(t_{ox})$ enter into anoxic environments. According to the above assumptions, $g_1$ will ultimately completely degrade under $O_2$-free conditions. However, the fate of $g_2$ is different: its degradation ceases. Burial efficiency—the fraction of organic matter delivered to the seafloor that survives subsequent degradation[16]—is then $g_2(t_{ox})/g_0 = ae^{-k_2 t_{ox}}$. Because burial efficiency decreases as the oxygen-exposure time $t_{ox}$ increases, there is a negative feedback: an increase in $O_2$ concentration leads to a longer oxygen-exposure time, more $O_2$ and organic matter are then consumed in degradation, and $O_2$ concentrations drop back to their initial levels[16–18]. However, this brings us to a conundrum: under the regulation of a negative feedback, how could $O_2$ concentrations have risen from the ancient low-level stable state to the modern high-level stable state? A positive feedback facilitated by POOM provides a possible resolution.

**Persistence of partially oxidized organic matter.** Organic matter degradation is a complex process involving physical, chemical, and biological reactions[15]. Microorganisms in sediments secrete enzymes to degrade organic matter, while the association of organic matter with minerals protects organic matter from enzymatic attack[19,20]. Organic matter adsorbed on mineral surfaces can further alter its own three-dimensional orientation via conformational changes and then form more condensed structures to resist microbial enzymes[18]. Incomplete degradation of organic matter in the presence of $O_2$ results in intermediate-stage metabolic products rich in reactive oxygen-containing functional groups such as carboxyls and hydroxyls[21–23]. These functional groups have been identified as characteristics of recalcitrant organic matter sampled from soil[24] and marine[25–27] environments. Laboratory experiments and field studies have demonstrated that reactive oxygen-containing functional groups have high adsorption energy and can enhance the strong association of organic matter with minerals such as iron oxides or clays[28,29], suggesting that reactive oxygen-containing functional groups in POOM promote mineral protection.

Figure 1 provides a conceptual picture. In panel (a), enzymes produced by microorganisms can access and more completely degrade organic matter, which is loosely bound to mineral surfaces in sediments. In panel (b), the partially oxidized organic matter is more tightly bound to mineral surfaces in sediments, limiting enzymatic access and inhibiting further degradation, which enhances the potential for the organic matter to persist on geologic timescales. A positive feedback, in which exposure to $O_2$

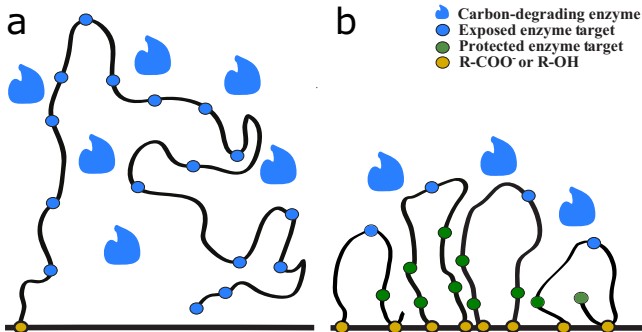

**Fig. 1 Comparison of biopolymers and their interaction with mineral surfaces before and after partial degradation by oxidative metabolisms.** Panel (**a**) illustrates a generic scenario in which only one site of the biopolymer is sorbed to the mineral surface (horizontal line) and the exposed enzyme targets on the biopolymer are freely accessible to carbon-degrading enzymes secreted by microorganisms. Panel (**b**) shows the result of further degradation of the biopolymer in (**a**) after partial oxidation. Reactive oxygen-containing functional groups, such as carboxyls (R-COO⁻) and hydroxyls (R-OH) are formed by oxidative enzymes (e.g., Baeyer–Villiger monooxygenases) that catalyze the production of partially oxidized organic matter in the presence of $O_2$. These functional groups create additional sorption sites, enhancing the association of the shorter organic carbon chains with the mineral surface. These partially oxidized, shorter organic carbon chains in (**b**) are more constrained compared to (**a**); consequently a large portion of enzyme targets on these shorter organic carbon chains are relatively inaccessible to microbial enzymes. Compared to (**a**), their degradation requires more investment of free energy to overcome the energy barrier that prevents enzyme access. The juxtaposition of (**a**) and (**b**) shows how partial oxidation impedes the biopolymer's accessibility to microbial enzymes and enhances its potential for long-term preservation.

leads to more organic burial and thus more $O_2$ accumulation, is therefore possible.

**Positive feedback in the ancient $O_2$-limiting sediments.** To specify the conditions under which this feedback works, we identify $g_2$ as POOM and extend Eqs. (1) and (2) to include a transformation of unoxidized organic matter ($g_1$) to POOM ($g_2$) with a rate constant $k_{12}$, as shown in Fig. 2a. The rate constant $k_{12}$ describes the integrated effect of oxidative metabolism and mineral protection on organic preservation. The model reads

$$\frac{dg_1}{dt} = -k_1 g_1 - k_{12} g_1, \qquad t \geq 0 \tag{4}$$

$$\frac{dg_2}{dt} = k_{12} g_1 - k_2 g_2, \qquad 0 \leq t \leq t_{ox}, \tag{5}$$

where $k_{12} = 0$ when $t > t_{ox}$, and with initial conditions again specified by (3). The solution for burial efficiency, $g_2(t_{ox})/g_0$, is derived in the "Methods" section and plotted for two different values of $k_{12}$ in Fig. 2b. A positive feedback (blue dashed curve in Fig. 2b) appears when $g_2(t_{ox})/g_0$ increases with $t_{ox}$ until it reaches its maximum at $t_{ox}^\star$ [Eq. (9)]. This increase of burial efficiency with increasing oxygen-exposure time occurs when

$$k_{12} > \frac{a}{1-a} k_2 \equiv k^\star. \tag{6}$$

This inequality states that when the POOM production rate constant $k_{12}$ is larger than POOM degradation rate constant $k_2$ modified by the factor $a/(1-a)$, the burial efficiency of organic matter increases with oxygen-exposure time up to the time $t_{ox}^\star$. In Earth's ancient $O_2$-limiting environment, organic matter not processed by $O_2$ is expected to dominate organic matter arriving

at the seafloor. This suggests $a \ll 1$, which in turn implies that the inequality in Eq. (6) would have been widely satisfied under $O_2$-limiting conditions. The positive feedback would eventually be taken over by a negative feedback (blue solid curve in Fig. 2b) when $t_{ox} > t_{ox}^\star$ (i.e., when the atmospheric $O_2$ reaches higher levels and the sedimentary environments become more oxygenated) because nearly all organic matter, protected by minerals or not, eventually degrades after long-term exposure to $O_2$. This negative feedback would stabilize $O_2$ at a new, higher-level stable state after Earth's oxygenation.

Enhanced burial of organic matter during Earth's oxygenation events has been attributed to an increase in the supply of nutrients, especially phosphorus (P), that promoted primary productivity[9,10]. The positive feedback mechanism described in this work instead implies that the elevation of $O_2$ levels derived from an increase in the burial efficiency of organic matter. This is supported by a recent study suggesting that burial efficiency rather than primary productivity was responsible for the substantial burial of organic matter in Earth's ancient $O_2$-limiting environment[30]. Moreover, our theory focuses on the carbon-oxygen system and assumes that the amount of organic-bound P buried with sinking organic matter remains unchanged, which implies that the C/P ratio of buried organic matter and therefore the net production of $O_2$ are not limited by the supply of P to the ocean.

Although the above analysis does not address the stability of the initial state, it nevertheless shows that early increases of $O_2$ levels in the ancient $O_2$-limiting environment can give rise to further increases via mechanisms of organic matter degradation and burial (i.e., physical protection of POOM) that operate in modern environments. But did the oxidative metabolisms producing POOM exist near the time of the GOE? We address this question next.

**Phylogenetic analyses.** The relevance of the POOM hypothesis to the rise of atmospheric $O_2$ crucially depends on the existence of POOM-producing oxidative metabolisms during Earth's oxygenation. One group of enzymes that can catalyze the formation of intermediate products in oxidative metabolisms are oxygenases[31]. Both laboratory experiments and field observations have shown that some oxygenases can insert one or two O atoms from $O_2$ into organic carbon chains to form reactive oxygen-containing functional groups[24,31]. Thus, reconstructing the evolutionary history of relevant oxygenases may provide a window through which we can investigate the possible role of POOM-producing oxidative metabolisms in Earth's oxygenation. The information from geologic records cannot directly reveal the evolution of enzymes, but phylogenetic methods can be used to reconstruct and analyze the history of enzyme families across geological timescales.

SAR202 bacteria (members of the phylum Chloroflexi) are ubiquitous in various modern environments and account for a significant fraction of the microbial population in the deep ocean[32,33]. They are especially prevalent in marine environments where recalcitrant organic matter is abundant[13,34]. Landry et al.[13] have demonstrated that SAR202 bacteria produce recalcitrant deep-ocean organic matter containing reactive oxygen-rich functional groups such as hydroxyls and carboxyls; moreover, they show that the Baeyer–Villiger monooxygenase (BVMO) enzymes play a predominant role in these oxidative metabolisms. BVMOs are a family of flavin-dependent oxygenases that can catalyze a wide variety of oxidation reactions of a large range of substrates[35]. Previous studies have suggested that some flavin-containing cofactors, including flavin adenine dinucleotide (FAD) and flavin mononucleotide (FMN) that are utilized by BVMOs, had already existed before the advent of the GOE[36] or even as old

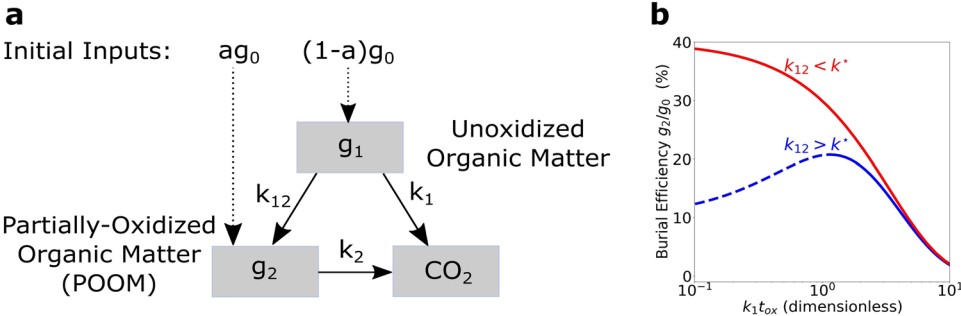

**Fig. 2 A theoretical prediction of a positive feedback responsible for Earth's oxygenation. a** Degradation paths of unoxidized organic matter and partially oxidized organic matter (POOM). The quantities $g_1$ and $g_2$ represent the amount of unoxidized organic matter and the amount of (physically protected) POOM deposited in sediments, respectively. Unoxidized organic matter is either directly oxidized to $CO_2$ with a rate constant $k_1$ or transformed to POOM with a rate constant $k_{12}$. POOM is oxidized to $CO_2$ with a rate constant $k_2$. **b** Burial efficiency $g_2/g_0$ as a function of dimensionless oxygen-exposure time $k_1 t_{ox}$. Positive feedback occurs when burial efficiency increases with oxygen-exposure time (blue dashed curve), which requires $k_{12} > k^\star$ [Eq. (6)].

as the age of the "RNA World"[37]. However, the evolution of BVMOs has rarely been explored. Here, we reconstruct the evolutionary history of BVMOs in the SAR202 bacteria and their closely related microbial species to test the hypothesis that POOM-producing oxidative metabolisms and Earth's oxygenation are temporally correlated.

To estimate the divergence and diversification times within the BVMO enzyme family and the SAR202 lineage, we reconstruct the phylogenies for 298 representative taxa (i.e., species tree) and 330 BVMO protein sequences encoded by genes within these taxa (i.e., gene tree), respectively. Details on sequence datasets, phylogenetic reconstruction, and molecular clocks and calibrations are provided in the "Methods" section and Supplementary Tables 1–3. The reconstructed trees and their chronograms are presented in Supplementary Figs. 1–4 and Supplementary Table 4. The horizontal gene transfer (HGT) events[38] of BVMO genes are inferred by reconciling the gene tree with the species tree ("Methods" section and Supplementary Figs. 5, 6).

HGT events can reflect adaptation to changing environmental conditions by acquiring new biological functions[38]. To investigate the relevance of these HGT events to the evolution of oxygen and carbon cycles, we construct the weighted distributions of the older and younger bounds for the timing of 68 HGT acquisitions of the BVMO gene that have bootstrap values ≥80% ("Methods" section). The age information of these HGT events is presented in Supplementary Table 5 and graphically summarized in Fig. 3. Supplementary Table 6 (in Supplementary Information) presents the directions (i.e., donors and recipients) of these HGT events, and Supplementary Fig. 7 graphically illustrates the directions of some representative (i.e., the oldest) HGT events.

Among the inferred HGT events (Fig. 3), the earliest acquisition of a BVMO gene (i.e., HGT event #1) occurred on the branch between the stem and crown SAR202 nodes (Fig. 3a). The older and younger bounds for the timing of this HGT event therefore correspond to the ages of the stem and crown SAR202 nodes, respectively. There is a 95% probability of this initial HGT into SAR202 occurring between 2600 Ma and 1540 Ma, with 85% and 97% of the probability density of the donor-recipient branch interval intersecting the GOE and Lomagundi Excursion Event, respectively (Fig. 3b). The Lomagundi Excursion Event, the largest known positive carbon isotope excursion event in Earth's history, has been interpreted as a consequence of a dramatic increase in the burial of organic carbon[39]. The temporal overlap of the initial HGT acquisition of BVMO gene with the GOE and Lomagundi Excursion Event suggests that there may exist some linkages between the diversification and ecological dispersal of POOM-producing genes and these significant geological events.

Figure 3 also shows that extensive HGT events of BVMO genes between/within the Chloroflexi, Actinobacteria, and Proteobacteria phyla (Supplementary Table 6) span the Proterozoic and Phanerozoic, apparently increasing in frequency starting in the Late Neoproterozoic. Although the extant taxa in the Actinobacteria and Proteobacteria phyla have not been demonstrated to be predominant in the production of POOM in Earth's modern environment, their ancestors might have been more important in this ecological role in the past when the $O_2$-limited marine environments were more extensive. However, the relatively large uncertainties in the dating of these HGT events prevent a more specific interpretation of whether and how this increasing trend of HGT frequency is related to the higher-resolution histories of oxygen and carbon cycles and other geological events, especially in the Phanerozoic, although such relations may exist. Future improvements in molecular clock methodologies may increase the precision of these estimates, enabling more detailed hypothesis testing.

Inset (a) of Fig. 3 indicates that, following the initial HGT acquisition, SAR202 BVMO genes diversified at rates that varied non-uniformly with time. Because we expect this variation to be coupled to Earth's redox state, we investigate it in detail in Fig. 4, which plots the per-gene diversification rate as a function of time ("Methods" section). We find three distinct bursts of diversification: during the Neoarchean/Paleoproterozoic (around 2500 Ma), the Middle/Late Mesoproterozoic (around 1200 Ma), and the Late Paleozoic/Early Mesozoic (around 300–200 Ma). The timing of these bursts correlates, respectively, with the GOE; the diversification of eukaryotic marine algae, especially major red and green algae lineages[40]; and the Permo-Carboniferous $O_2$ pulse. In addition, a large drop in the diversification rate occurred during the period that includes the Neoproterozoic global glaciations[41].

To assess the significance of these fluctuations, we constructed and analyzed a null model in which genes diversify at random times at the average rate of our reconstructed tree ("Methods" section). On average, the number of nodes in the null model grows exponentially with time, but the fluctuations from the average are random and uncorrelated in time; that is, they are white noise. In contrast, the fluctuations of the real data are highly correlated in time and are significantly different than white noise (Supplementary Fig. 8). These results demonstrate that the temporal structure of the diversification rates in Fig. 4 is inconsistent with the expected rates under a null evolutionary model, and likely includes time intervals of substantial environmental perturbation; we propose that these perturbations reflect, at least in part, the influence of Earth's evolving redox state.

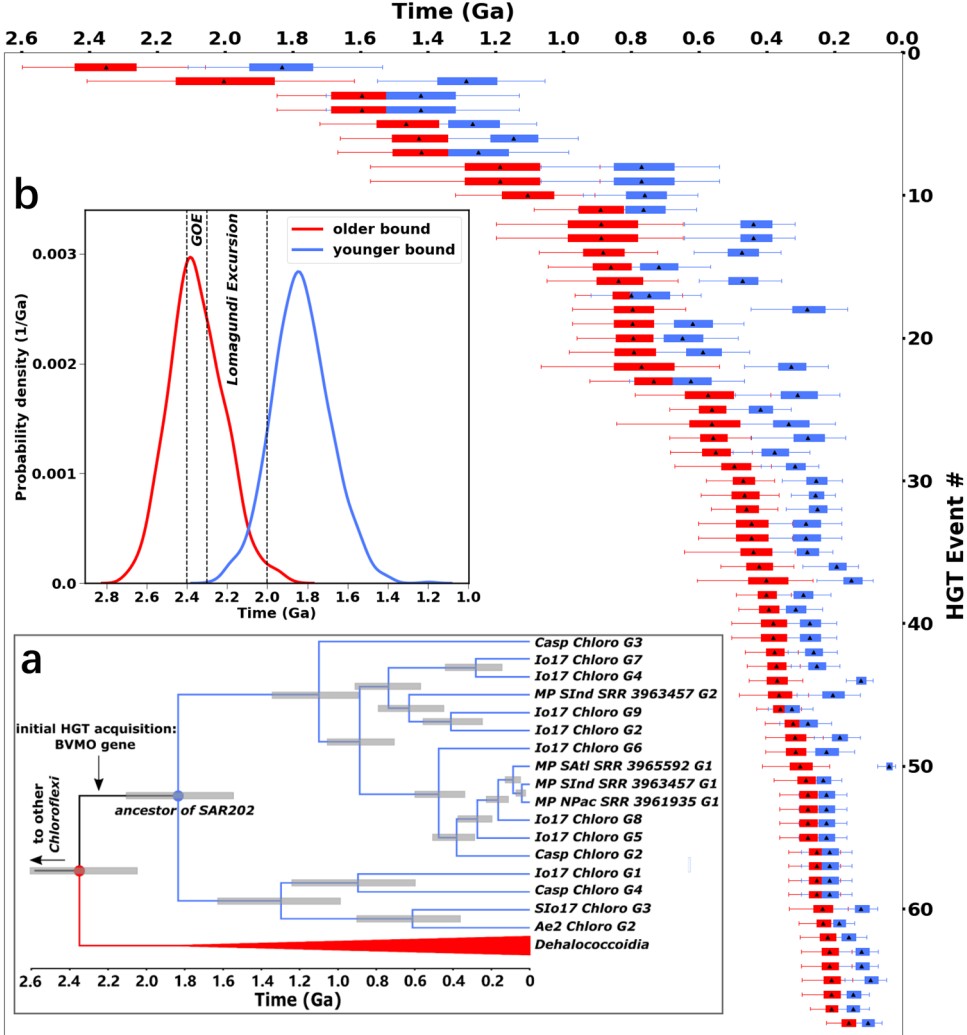

**Fig. 3 Phylogenetic and molecular clock analyses for the Baeyer–Villiger monooxygenases (BVMOs) of the SAR202 cluster bacteria and their closely related microbial species.** The main figure shows a graphic summary for weighted means and 95% confidence intervals (CIs) of the older and younger time bounds ($n = 1023$ posterior samples for the BVMO gene tree chronogram) for 68 inferred horizontal gene transfer (HGT) events between/within the Chloroflexi, Actinobacteria, and Proteobacteria phyla presented in Supplementary Information, Supplementary Table 5. The directions (i.e., donors and recipients) of these HGT events are provided in Supplementary Table 6. In the main figure, red and blue boxes represent older and younger time bounds, respectively. The right and left bounds of each box are the 25th and 75th percentiles, respectively; the right and left whiskers mark the 2.5th and 97.5th percentiles, respectively; the black triangles represent means. **a** A subtree of calibrated chronogram showing SAR202 (blue) and related Dehalococcoidia group (red); the complete chronogram is provided in Supplementary Fig. 4. The initial HGT acquisition (also illustrated in Supplementary Fig. 7) occurred on the branch between stem SAR202 node (red filled circle) and crown node SAR202 (blue filled circle). Gray horizontal bars on the nodes indicate 95% CIs ($n = 1023$ posterior samples for the BVMO gene tree chronogram). **b** The posterior date intervals of the older and younger age bounds for the initial HGT event into SAR202 (i.e., the stem (red) and crown (blue) SAR202 nodes in (**a**)). Stem and crown date intervals correspond to the distributions of older (red) and younger (blue) time bounds for the initial HGT acquisition shown in (**a**), and also correspond to the HGT event #1 shown in the main figure. The mean date of older bound is 2350 Ma (95% CI: 2056 Ma–2598 Ma), and the mean date of younger bound is 1830 Ma (95% CI: 1535 Ma–2110 Ma), where $n = 1023$ posterior samples for the BVMO gene tree chronogram. The time windows of the Great Oxidation Event and the Lomagundi Excursion Event overlap the distributions of older and younger time bounds.

## Discussion

The POOM hypothesis suggests that the interactions of oxidative metabolites with sedimentary minerals enhance organic matter burial in low-O$_2$ (but not anoxic) environments. Oxygen-dependent enzymes, such as those which create POOM, likely diversified as oxygenated environments themselves became more diverse. Therefore, changes in O$_2$ levels associated with organic matter burial should correlate with the ecological diversification of oxidative metabolisms.

In the Late Archean, initially low and localized O$_2$ production likely instigated the diversification of aerobic metabolic pathways, such as the POOM-producing oxidative metabolisms. This is supported by the early divergence of BVMO genes in different microbial lineages (Supplementary Figs. 3 and 4). The acquisition of the BVMO gene via an HGT event (Fig. 3a) may represent adaptation to the new, weakly oxygenated environment in the Neoarchean/Paleoproterozoic. BVMO genes than more rapidly diversified (as seen in the prominent peak at 2500 Ma in Fig. 4a) and transferred across environments, which then enhanced the production of POOM. Furthermore, with the rise of atmospheric O$_2$, iron(II) dissolved in seawater or contained in minerals such as pyrite (FeS$_2$)[42] was oxidized to iron(III)[43], potentially promoting the physical protection of organic matter[18,19]. As a result, POOM produced by the oxidative metabolisms would have been

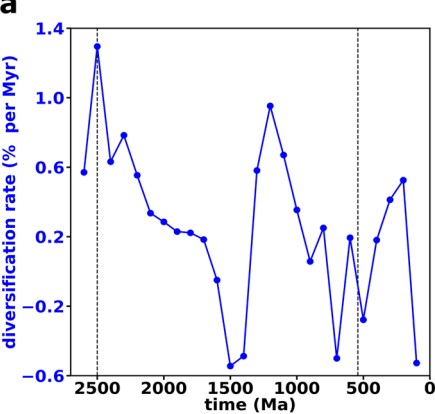

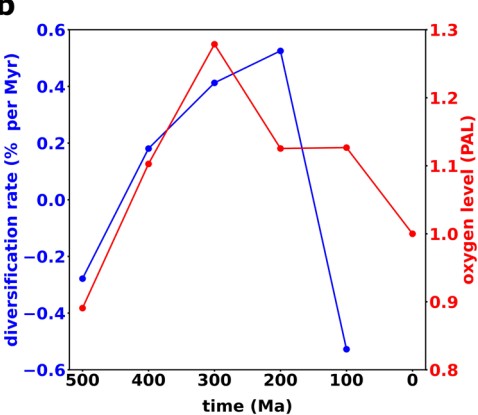

**Fig. 4 Diversification of SAR202 Baeyer–Villiger monooxygenase (BVMO) genes and its temporal correlation with the evolution of Earth's atmospheric O₂ levels. a** Evolution of the per-gene diversification rate of the SAR202 BVMO genes over geologic time. The two vertical dashed lines represent the Archean/Proterozoic boundary and Proterozoic/Phanerozoic boundary, respectively. Three positive peaks stand out, during the Neoarchean/Paleoproterozoic (around 2500 Ma), the Middle/Late Mesoproterozoic (around 1200 Ma), and the Late Paleozoic/Early Mesozoic (around 300–200 Ma), corresponding, respectively, to the time of the Great Oxidation Event, the rapid divergence of eukaryotic marine algae, and the Permo-Carboniferous O₂ pulse. **b** The temporal correlation between the SAR202 BVMO diversification rate (blue) and Earth's atmospheric O₂ level (red) in the Phanerozoic. The red dots from 100 Ma to 500 Ma are the moving averages of the O₂ levels in[45], using an averaging window of 200 Myr, which is the uncertainty range (i.e., 95% confidence interval) of the age distributions in the Phanerozoic on the gene tree chronogram of BVMO homologs (Supplementary Fig. 4). The O₂ concentration at time 0 is the present atmospheric level (PAL).

strongly protected by the newly accumulated iron(III) minerals, thereby enhancing the accumulation of atmospheric O₂. The combination of these factors explains the correlation of the initial diversification of the SAR202 BVMO genes with the GOE and the Lomagundi Excursion Event.

Oxidative metabolisms likely further spread during the Phanerozoic, promoting POOM formation and O₂ accumulation. This connection may have been facilitated by the biominerals[43] and clay minerals[43,44] that had accumulated on Earth's surface since the Early Phanerozoic. Such facilitation should have been caused by the increase in the abundance of clay minerals but does not depend on their specific trends (i.e., whether a primary trend or a diagenetic/metamorphic trend). The Phanerozoic diversification rate reached a maximum in the Permo-Carboniferous, when burial rates were likely elevated and atmospheric O₂ levels

are thought to have been extraordinarily high[45]. Moreover, diversification rates appear to broadly track the evolution of Phanerozoic O₂ levels (Fig. 4b).

Figure 4 also shows that BVMOs rapidly diversified in the Middle/Late Mesoproterozoic (around 1200 Ma). While atmospheric O₂ levels remained low, the diversification of eukaryotic marine algae, such as red and green algae, was a key biological feature of Proterozoic oceans[40]. The resistant biopolymers in the cell walls and cysts of these organisms likely enhanced organic matter burial and therefore O₂ accumulation[46]. The relatively high diversification rates at this time may result from the expansion of oxidative metabolisms in the oxygenated niches. If so, the apparent environmental stasis during this so-called "boring billion"-year period may be a reflection of scarce research and scant geochemical evidence[47]. The possibility of one or more local oxygenation events[47,48] during this time merits further investigation.

Anoxic conditions and low primary productivity likely prevailed beneath extensive ice cover during the Neoproterozoic glaciations[41]. In such environments, our hypothesis predicts that diversification of BVMO genes would stagnate, which potentially explains the steep decline around 700 Ma in Fig. 4a. Overall, the temporal correlations between the evolutionary histories of the BVMO diversification and atmospheric O₂ levels suggest not only that environmental oxygenation is a driver of the evolution of this gene family but also that but also that oxidative metabolism catalyzed the early rise of O₂, thereby providing insight into poorly understood stages in the history of Earth's redox state.

The results presented here suggest several avenues for further investigation of the POOM hypothesis. BVMOs in SAR202 are an important enzyme family for the production of POOM in modern marine environments, and have a history that can be traced back across the Phanaerozoic and Proterozoic Eons, showing a remarkable degree of phyletic fidelity for a gene family this ancient. For these reasons, the history of BVMOs is especially useful for testing the POOM hypothesis. However, single-gene families are inherently limited in the temporal precision that can be attained in using molecular clock dating methods; examining the evolutionary history of other similarly suitable gene families that generate POOM in these environments could therefore potentially provide further support for our hypothesis. Moreover, the ancestors of the Actinobacteria and Proteobacteria phyla are expected to have contributed to the formation of POOM in deep time (as discussed in the Phylogenetic Analyses section); further confirmation of this speculation by laboratory/field investigations would support the POOM hypothesis. Besides, additional laboratory studies of the preservation potential of POOM under conditions analogous to Earth's ancient low-O₂ marine environments could also provide valuable experimental validation. Finally, geochemical analyses of the variation of (mineral-associated) POOM abundance in sedimentary records around the time of Earth's oxygenation events or oceanic anoxic events[49], possibly employing ramped pyrolysis/oxidation[20], may provide additional opportunities for testing the POOM hypothesis.

In summary, the POOM hypothesis provides a positive-feedback mechanism for Earth's oxygenation deriving from the interactions of oxidative metabolic products with sedimentary minerals. The temporal connections between the evolution of a representative oxidative enzyme family, BVMOs, and the evolution of Earth's atmospheric O₂ supports this hypothesis. Counterintuitively, the expansion of oxidative metabolisms coupled to the evolution of sedimentary minerals created an autocatalytic dynamics in which the initial advent of oxygen-consuming processes led to oxygen accumulation. The complexity of biogeochemical cycles suggests that similar synergies may help explain

other aspects of the episodic character of Earth's biogeochemical evolution.

## Methods

**Positive feedback deriving from partially oxidized organic matter.** As indicated in Fig. 2a, the unoxidized component $g_1$ of organic carbon will eventually completely degrade regardless of the presence of $O_2$; thus it does not contribute to organic matter burial. However, the degradation of the POOM component, $g_2$, ceases in the absence of $O_2$. The burial efficiency is therefore proportional to $g_2(t_{ox})$, which is the quantity of POOM remaining when exposure to oxygen terminates at time $t = t_{ox}$. Solving Eqs. (4) and (5) with the initial conditions (3) then provides an exact expression for the burial efficiency:

$$\frac{g_2(t_{ox})}{g_0} = \frac{\left[(1-a)e^{(k_2-k_{12}-k_1)t_{ox}} - 1\right]k_{12} + a(k_2-k_1)}{(k_2-k_1-k_{12})e^{k_2 t_{ox}}}. \quad (7)$$

A positive feedback occurs at times when $dg_2/dt > 0$. This condition may be satisfied only if the rate constant $k_{12}$ for the conversion of $g_1$ to $g_2$ is sufficiently fast. The requirement that $dg_2/dt > 0$ imposes the sharp lower bound

$$k_{12} > \frac{a}{1-a}k_2 \equiv k^\star. \quad (8)$$

When this inequality holds, the maximum of $g_2$ occurs at the critical oxygen-exposure time $t_{ox}^\star$ where $dg_2/dt = 0$. Using Eq. (7), we find

$$t_{ox}^\star \equiv \frac{\log\left[\frac{k_2(ak_2 - ak_1 - k_{12})}{(a-1)k_{12}(k_{12}+k_1)}\right]}{k_2 - k_1 - k_{12}}. \quad (9)$$

Thus the positive feedback appears when $k_{12} > k^\star$ and $t < t_{ox}^\star$.

**Species tree and gene tree reconstruction.** Protein Basic Local Alignment Search Tool (BLASTp) on the National Center for Biotechnology Information (NCBI) database was used to search the proteins of interest. For the species tree reconstruction, we used the sequences that are homologous to 30 ribosomal proteins (Supplementary Table 1) of *SAR202 cluster bacterium* Io17-Chloro-G4. Ribosomal proteins are from 298 taxa (Supplementary Table 2), including 203 taxa directly collected from the NCBI database and 95 taxa provided in[50]. To reconstruct the gene tree of BVMOs, we used 330 protein sequences homologous to the BVMO of *SAR202 cluster bacterium* Io17-Chloro-G4 (NCBI Query ID: PKB68843.1 [https://www.ncbi.nlm.nih.gov/assembly/GCA_002816455]); 31 of these BVMOs belong to the SAR202 cluster. To check the homology of those oxygenase genes that were not well annotated in the original database, we employed the method used in[13] and ran the Phyre2 structural homology recognition server[51] with the "normal modeling mode". It turned out that the query sequences were modeled with 100% confidence and high coverage (≥85%) by the BVMO template in the Phyre2 system, suggesting that those sequences have high predicted structural similarity to BVMOs. A typical BVMO model generated by the Phyre2 server with the "normal modeling mode" is provided in Supplementary Fig. 9. The sequences were aligned using the MAFFT program[52] with the progressive method "FFT-NS-2" and the score matrix "bl 62" (i.e., BLOSUM62). Alignments were visualized on Clustal X[53], and poorly aligned regions, which primarily consist of highly variable C-terminal regions, were manually deleted. The aligned ribosomal protein sequences were concatenated using the program SequenceMatrix[54]. The ribosomal protein and BVMO protein sequence alignments are provided in Supplementary Data 1 and 2, respectively.

Both the maximum-likelihood species tree and gene tree were inferred using IQ-TREE (version 1.6.3)[55] with the ultrafast bootstrap algorithm[56] and 100 replicates. The ModelFinder Plus option was used in IQ-TREE to identify the best-fit substitution model for each sequence dataset. The species tree alignment best fit the "LG+I+R9" model, using an LG model for amino-acid exchange rate matrices[57], invariable sites, and a Free Site Rate model with 9 categories of rates[57]. The "LG+R6" model best fit the BMVO protein tree, using an LG model with 6 Free Site Rate categories.

Previous studies and outgroup rootings have consistently supported a species tree rooting of Bacteria where Proteobacteria, Bacteroidetes, Ignavibacteria and Chlorobi are grouped in one clade, while Actinobacteria, Chloroflexi and Cyanobacteria are grouped in the other clade[58]. Therefore, we manually rooted the species tree based on the above information. Since we have no clear knowledge about the root for the gene tree of BVMOs, the minimal ancestor deviation (MAD) method[59] was used to infer its root position. The rooted species tree and gene tree are in Supplementary Data 3 and 4, respectively.

**Divergence time estimation for species tree.** The divergence time was inferred using the program PhyloBayes (Version 4.1c)[60] with parameters "-catfix C60 -ugam -bd -sb -nchain 2 100 0.3 50" and all other parameters default. The timing of the earliest evidence for Earth's habitability—4.4 Ga zircon[61]—was used as the older (i.e., lower) bound for the root prior, while the timing of the earliest known bacterial microfossils – 3.4 Ga stromatolites in Warrawoona Group, Australia[62] was used as the younger (i.e., upper) bound for the root prior. Since our belief in the true root age becomes weaker as the date approaches the two bounds (4.4 Ga and 3.4 Ga), we approximated this by using a normally distributed root prior with a 95% confidence interval across the range

(that is, a mean of 3.9 Ga and a standard deviation of 0.25 Ga). The uncorrelated gamma model (i.e., -ugam)[63] was used for the relaxed molecular clock. Since the estimated oldest habitability was 4.4 Ga[61] and we have a strong belief that the node of last universal common ancestor (LUCA) should be deeper than the root node on the species tree[64], a hard bound was imposed on the older age of the root prior in the calibration file. However, we set a soft bound[65] on younger age of the root prior. While it is unlikely that the 3.4 Ga stromatolites represent stem bacterial diversity[50], it remains a possibility, and therefore the model should permit the root age of the species tree to be younger than 3.4 Ga.

The 95% confidence intervals of posterior ages on following crown and stem nodes obtained in a previous study[50] were used as the secondary calibrations for the species tree: (1) Crown Chloroflexi_ 3.374 Ga–2.674 Ga; (2) Stem Chloroflexia: 2.782 Ga–2.053 Ga; (3) Crown Chloroflexia: 2.041 Ga–1.645 Ga; (4) the common ancestor of Ktedonobacteria and Dehalococcoidia: 2.995 Ga–1.793 Ga; (5) Crown Dehalococcoidia: 2.013 Ga–0.789 Ga; (6) the common ancestor of Anaerolineae and Chloroflexia: 3.124 Ga–2.384 Ga. These uniform age priors have broad ranges and therefore are unlikely over-specifying due to some bias from previous analyses[50], which is appropriate because our knowledge of the divergence time on these nodes is limited.

The chronogram of species tree was generated using the "-readdiv" command in PhyloBayes[60]. Since PhyloBayes is a phylogenetic Monte Carlo Markov Chain (MCMC) sampler[60], the sampled values in the initial phase are likely to be outside a high probability region[66]. To exclude the influence of the starting point on chronogram computation, the first 20% of sampled node ages were discarded (i.e., "burned in"). The species tree chronogram is in Supplementary Data 5. The command "-v" in PhyloBayes was used to output all dated trees of the samples in the "datedist" file, which were used to calculate the node age distributions and the divergence rates of genes. To compare posterior ages to estimations obtained without the information from gene sequences, we generated the age estimates using "-prior" command in PhyloBayes.

**Divergence time estimation for gene tree.** The 95% confidence intervals for five nodes in the SAR202 clade on the species tree were used as the secondary calibrations to estimate the divergence time on the gene tree. These SAR202 nodes in the species tree all have high bootstrap supports (>90%) and are also present in the gene tree. It has been suggested that the diversification time of oxygen-consuming metabolisms should be roughly consistent with the timing of GOE[67]. However, due to the lack of prior knowledge about the root of the gene tree, we imposed a generous flat root prior of 3.2 Ga–1.2 Ga to avoid false precision. The older bound of this range (i.e., 3.2 Ga) not only corresponds to the published age estimates[50] for stem Cyanobacteria (i.e., the split between Cyanobacteria and non-photosynthetic lineages), but also is older than nearly all proposed geochemical evidence for oxygen[1,2]. The younger bound was set as 1.2 Ga, which is sufficiently young to include the 95% confidence interval for the age of the crown node of the SAR202 clade on the species tree. The gene tree chronogram is in Supplementary Data 6.

**Horizontal gene transfer inference.** Phylogenetic reconciliation was performed using the program RANGER-DTL (Version 2.0)[68]. The cost scores of duplication, transfer, and loss (DTL) were set as default values (i.e., 2, 3, and 1). The rooted gene tree was mapped to the rooted species tree to infer the horizontal gene transfer (HGT) events occurred in the evolutionary history of BVMOs. The mapping was repeated 100 times, and the estimated HGT events that have high bootstrap supports (>80%) were selected for further analyses. These HGT events are within or between the Chloroflexi, Actinobacteria, and Proteobacteria phyla.

The age distributions of the recipients for these HGT events were computed using the "datedist" file of species tree and the following rules: (i) for transfers that have multiple (more than one) non-leaf recipients, we computed the fraction of each recipient as its weight and used these weights to calculate the weighted age distribution; (ii) for HGTs which have one or more than one leaf recipients, we deleted the leaf recipients, only kept the "internal node" recipients, computed the fraction of each recipient as its weight and used these weights to calculate the weighted age distribution; (iii) for HGTs that have one or more than one leaf donors, we included the ages of the recipients for these transfers when constructing the age distributions. These weighted age distributions of recipients show the ranges of the younger bounds for the HGT events.

**Divergence rates of BVMO genes.** We calculated the diversification rates of SAR202 BVMO genes using their posterior age data. We seek the per-gene diversification rate $r$, which in continuous time would be defined by

$$r = \frac{1}{N}\frac{dN}{dt} = \frac{d\log N}{dt}, \quad (10)$$

where $N(t)$ is the number of nodes at time $t$ after the first 20% of the trees in the "datedist" file were "burned-in". To compute $r$ in discrete time, we denote the average number of nodes at time $t_i$ by $N_i$, where the bin width $\Delta t = t_{i+1} - t_i = 100$ Myr. Then the diversification rate $r_i$ at time $t_i + \Delta t/2$ is given by

$$r_i = \frac{1}{\Delta t}\log\frac{N_{i+1}}{N_i}. \quad (11)$$

Figure 4a presents the time series of $r_i$.

To assess the significance of the fluctuations of $r_i$, we consider a null model in which a gene tree randomly diverges at each time step with a constant probability that equals the average diversification rate $\bar{r}$ in Fig. 4a. On average, the number of nodes in the null model grows exponentially with time, like $e^{\bar{r}t}$. However the fluctuations

$$x_i = \log \frac{N_{i+1}}{N_i} - \bar{r}(t_i + \Delta t/2) \qquad (12)$$

are random and uncorrelated—i.e., they are white noise, which we verified numerically. In contrast, calculations of the fluctuations $x_i$ from the SAR202 BVMO diversification rates yields a highly correlated time series, as can be seen in Fig. 4a. To quantify this difference, we computed the power spectrum (i.e., the amplitude of the Fourier transform, squared) of $x_i$, which is shown in Supplementary Fig. 8. Our numerical analysis shows that the probability that the white noise of the null model would generate the highest peak in Supplementary Fig. 8 is $p < 0.01$. Moreover, the $p$-values for a Ljung–Box white noise test[69] at 25 lags are all greater than 0.5. Thus the fluctuations of the diversification rates in Fig. 4 are statistically distinct from the results which would be obtained from a randomly branching gene tree.

**Reporting summary**. Further information on research design is available in the Nature Research Reporting Summary linked to this article.

## Data availability

All data needed to evaluate the conclusions in the paper are presented in the main text and/or the Supplementary Information. The information of databases, accession numbers, and hyperlinks for the sequence data of all taxa used in this study is provided in Supplementary Table 2. All sequencing data used in this study are provided in Supplementary Data 1 and 2. All maximum-likelihood trees and chronograms generated in this study are provided in Supplementary Data 3–6. These data are also available online (https://doi.org/10.5281/zenodo.5914862)[70]. Source data are provided with this paper.

## Code availability

Code used in this study is available online (https://doi.org/10.5281/zenodo.5914862)[70].

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

## Acknowledgements

The authors thank T. Bosak, Y. Cohen, and D. Repeta for insightful discussions, Z. Landry and S. Giovannoni for providing us the gene sequences of SAR202 flavin-dependent monooxygenase, and D. Colatriano and D. Walsh for providing us the raw gene sequences of SAR202 16S rRNA. This work was supported in part by the mTerra Catalyst Fund (DHR), NSF Award EAR-1338810 (DHR), and NSF Integrated Earth Systems Program Award 1615426 (GPF).

## Author contributions

H.S., D.H.R., and G.P.F. designed research, performed research, and wrote the manuscript.

## Competing interests

The authors declare no competing interests.
