## [Peer Review File · Nature Communications]

Oxidative metabolisms catalyzed Earth's oxygenationReviewers' Comments:

Reviewer #1:

Remarks to the Author:

This manuscript presents the interesting hypothesis that the step increase in atmospheric oxygen level, from essentially zero to a level sufficient to suppress the preservation of the mass-independent fractionation of sulfur isotopes (around 1/100,000 of the present level) was facilitated by positive feedback arising from the production of partially oxidized organic matter. The feedback involves the higher preservation potential of POOM because of its stronger adsorption onto minerals. The authors propose to substantiate this hypothesis through phylogenetic analysis of the oxygenates (required to form POOM) that is broadly consistent with our understanding of the oxygenation history of the planet.

I find the argument interesting, if not particularly compelling. I think it could be strengthened, and provide some comments below toward that end.

Line 24-25: I don't think we should present the evolution of atmospheric O₂ in this fashion. Given the short residence time of O₂ in the atmosphere (even today it is ca. 1 million years), the evolution of its atmospheric level has been through a series of quasi-steady states. Yes, a slight excess of production over consumption drives these transitions, but that excess can be very small and must be strongly regulated. The standard story is that in the Archean, the potential consumption of O₂ exceeded the biota's capacity to produce O₂ through overall net oxygenic photosynthesis (burial of organic carbon or its proxy in other reduced substances such as pyrite or reduced iron minerals). I say potential, because even in the Archean the production and consumption of O₂ had to be closely balanced. Then near the Archean/Proterozoic boundary, something changed to allow for a transition to an oxygenated atmosphere. Either the potential consumption diminished below the production rate, or the production rate increased to exceed that potential consumption. Then atmospheric O₂ rose until again consumption again rose to meet (perhaps elevated rates of) production (e.g., by activating new sinks for O₂ associated with the oxidative weathering of reduced crustal materials, a process that requires higher O₂ than earlier sinks like volcanic gas oxidation). Alternatively, here, O₂ levels rose until the burial efficiency increased via POOM stabilizing O₂ production at a new, higher level that presumably was then matched by a higher rate of consumption (all requiring higher O₂ levels).

Line 27: so yes, a switch from one steady state (essentially zero) to a new one. I don't agree though that this requires positive feedback; a temporal decline in the potential consumption rate (e.g., a decline in the volcanic sink) could eventually reduce the sink below the source, allowing O₂ to rise until a new sink was established (as above, e.g., through oxidative weathering).

Line 54: If g₁ ultimately completely degrades under O₂-free conditions, then doesn't this mean that there would have been no organic matter burial in the Archean?

Line 57: The whole argument of this manuscript is built on the tacit assumption that nutrient supply (e.g., of phosphate) doesn't limit the burial rate of organic matter. That should be stated explicitly.

Line 68: somewhere the authors need to address the claim by Kennedy et al. (2006; Science 311(5766):1446-9) that the types of clay minerals the authors invoke here to be strongly adsorbing POOM didn't appear in the environment until nearly 2 billion years after the Great Oxidation Event, and thus weren't available to support the hypothesized mechanism presented here. I think Kennedy et al. fundamentally misinterpreted the observed trends in clay mineral abundance through time as a primary rather than a diagenetic / metamorphic trend, but to my knowledge no one has ever published a challenge to that paper. I think the current authors need to do so.

Line 71: POOM is partially oxidized by definition. That means, per unit C buried, it is less effective at net O₂ production. I think this needs to be considered / quantified by the authors. Without the

enhanced burial efficiency of POOM argued here, the shift from the burial of non-oxidized organic matter in the Archean to POOM in the Proterozoic would have REDUCED oxygen production! Here again, consideration of C/P ratios along with the oxidation state of the C being buried would be helpful.

Line 89: remove "be" (and by the way, on line 6 and elsewhere, ...ly adverbs don't take hyphens)

Line 94: this statement of the positive feedback could be reversed to say "a decrease in oxygen level can give rise to further decreases via" the same mechanism. Yet the oxidation of the atmosphere apparently was irreversible. I think the authors need to address the reversibility of their positive feedback vs. the irreversibility of atmospheric oxygenation.

(I'm afraid I have no expertise to comment on the phylogenetic analyses)

Line 140: "excursion"

Lee Kump, Penn State

Reviewer #2:

Remarks to the Author:

The manuscript on Oxidative metabolisms catalyzed Earth's oxygenation by Shang, Rothman and Fournier deals with an important part of the evolutionary history known as Great Oxidation Event. Although the presented hypothesis is quite interesting it needs more support and further complex & comprehensive considerations in comparison with published alternative explanations of this evolutionary event on Earth.

From these 16 points it is clear that I suggest MAJOR REVISION of this manuscript to be considered and I can review it again after improvements.

These are important points that authors need to consider for preparation of an improved version of this manuscript:

1. Immediately at beginning of the abstract, lines 3-4:

"GOE is generally explained by the burial of reduced organic matter..."

-but it is primarily explained by different isotopic signatures in rocks older and younger than 2.4 billion years suggesting a significant difference in the content of Earth's atmosphere (see e.g. Nature Vol. 443 (2006) pp. 643-645 and references therein.). Only secondarily the proposed organic-carbon burial would trigger GOE. This needs to be explained more clearly for those readers that are not deeply involved in this topic.

2. Abstract lines 9-10 authors need to present here clearly which is this "one key enzyme family that generates POOM in marine systems". This is essential for topical searches of potentially interested readers.

3. Line 38 on page 2: "In modern sediments, POOM is produced by aerobic metabolisms with the aid of oxygenases" this is claimed without any citation. So are oxygenases just to aid this process or are they indeed key enzymes that allow production of POOM?? Could potentially other enzymes besides oxygenases also be involved in this process?

4. Lines 40-41: authors selected Baeyer-Villiger monooxygenase for further work.

It needs to be added here that they are flavin-dependent enzymes. But why not mention also some alternatives e.g. P450 monooxygenases that are heme-dependent and are also expected to have very

ancient origin. Could alternative oxygenases not contribute to this important process just "BVMO"?

5. Line 52 on page 3: "labile component g1 and recalcitrant component g2" - this seems a very simplified scheme but the most important point of criticism here is that - most organic components do not decay in the presence of molecular O₂ easily. This mathematical model therefore appears to be a comfortable simplification with respect to expected real conditions in primordial Earth surface. Molecular oxygen is not so reactive itself but there are the so called reactive oxygen species ("ROS") i.e. hydrogen peroxide, superoxide radical, hydroxyl radical, hydroperoxyl radical, nitric oxide radical and last but not least singlet oxygen - all products of metabolism, or present in the environment that react with organic components readily.

6. Line 60 on page 4: when writing about conundrum, it is even much more complicated if we consider point 5. above. references #15 & #16 are quite old and partially speculative. Besides the effect of POOM some other factors derived e.g. from the presence of reactive oxygen species can have had influence on particular (reactive) oxygen concentrations that were probably not identical and homogenous over the whole Earth's surface.

7. line 66: this is a very general statement "secrete enzymes to degrade organic matter". Namely, some enzymes like hydrolases do not need to use oxygen to make their efficient reaction of organic matter degradation.

8. Figure 1 on page 5 - if really conceptual this figure certainly needs an upgrade. What kind of enzymes are considerable for this scheme? Divergent oxygen-containing functional groups need to be clearly labelled and distinguished here otherwise this figure looks just like a sketch from an undergraduate-level book.

9. Line 101 - Reference #30 from the year 1992 is really very old. Was there no published progress in the meantime? Most oxygenases are either flavin or heme dependent. Was the availability of these essential cofactors before and during GOE for the proposed POOM formation guaranteed as well? This could be followed with the parallel phylogeny of flavin and heme producing pathways (or at least discussed shortly).

10. Line 125 and Supplementary Figures S1-S4. The fonts used in supplementary figures are so small that they are really hardly readable even under a huge magnification. In some cases they are even overlapping. Most readers will be impatient to study for hours the important details of these evolutionary figures. Thus the claimed HGT events are not really obvious from the current presentation of these results (e.g. HGT event #1 described in line 134). Authors need to label clearly the directions of HGT between various bacterial phyla. Moreover, Table S5 just presents some numbered nodes for respective donors and recipients. This is almost impossible to follow and resume for an average reader. It shall be clearly declared - beyond the numbered nodes in this table - if a particular HGT occurred e.g. from Cyanobacteria to Chloroflexi etc.

11. Figure 4 on page 11 - the authors mention also a rapid divergence of Eukaryotes that reveals also some differences in timing in various literature sources. Can the evolution of Eukaryotes and their smarter metabolism have also an significant influence & impact on the atmospheric oxygen level and the production of POOM?

12. Lines 133 onwards - the presentation of observed HGT events is dominantly focused on SAR202 bacteria. But from the first glance of presented evolutionary tree it is obvious that BVMO genes are present also in other (important) bacterial phyla namely: Cyanobacteria, Proteobacteria, Actinobacteria, Bacteroidetes to name just few of them. Could have more divergent bacterial phyla beyond SAR202_Chloroflexi lineage also contributed to the global formation of POOM?? At least some more comprehensive evaluation in this respect is needed here.

13. Lines 179-181 on page 12: "emergence of aerobic metabolic pathways" this is just a very general declaration. It is desirable that authors confront diverse bacterial phyla (mentioned in point 12.) and their proposed ancestors at the estimated time of "Late Archean" to discuss their respective capacity in such aerobic metabolic pathways.

14. Lines 185-186 on page 13: Iron II was oxidized to Iron III. In context with ROS mentioned in point 5. - if just traces of peroxide were present around the ancestral POOM producing cells, then the so called Fenton reaction could have occurred with some consequences that would not be as protective as declared here.

15. Lines 252-253 on page 16: "they have high predicted structural similarity" - this needs some quantification or more exact description of critical parameters to be able to repeat this procedure on Phyre2 or alternative homology modelling servers. And what were the optimal (optimized) parameters for the alignments? Eventually a typical Phyre model could be presented as supplementary material.

16. Line 255 "were concatnated" shall be written as concatenated. This is just a small typing error but the option concatenated sequences could alternatively be used also for multiple oxygenases...

Reviewer #1 (Remarks to the Author):

This manuscript presents the interesting hypothesis that the step increase in atmospheric oxygen level, from essentially zero to a level sufficient to suppress the preservation of the mass-independent fractionation of sulfur isotopes (around 1/100,000 of the present level) was facilitated by positive feedback arising from the production of partially oxidized organic matter. The feedback involves the higher preservation potential of POOM because of its stronger adsorption onto minerals. The authors propose to substantiate this hypothesis through phylogenetic analysis of the oxygenates (required to form POOM) that is broadly consistent with our understanding of the oxygenation history of the planet.

I find the argument interesting, if not particularly compelling. I think it could be strengthened, and provide some comments below toward that end.

Authors' Response:

Thank you very much for your thoughtful and constructive comments.

Line 24-25: I don't think we should present the evolution of atmospheric O₂ in this fashion. Given the short residence time of O₂ in the atmosphere (even today it is ca. 1 million years), the evolution of its atmospheric level has been through a series of quasi-steady states. Yes, a slight excess of production over consumption drives these transitions, but that excess can be very small and must be strongly regulated. The standard story is that in the Archean, the potential consumption of O₂ exceeded the biota's capacity to produce O₂ through overall net oxygenic photosynthesis (burial of organic carbon or its proxy in other reduced substances such as pyrite or reduced iron minerals). I say potential, because even in the Archean the production and consumption of O₂ had to be closely balanced. Then near the Archean/Proterozoic boundary, something changed to allow for a transition to an oxygenated atmosphere. Either the potential consumption diminished below the production rate, or the production rate increased to exceed that potential consumption. Then atmospheric O₂ rose until again consumption again rose to meet (perhaps elevated rates of) production (e.g., by activating new sinks for O₂ associated with the oxidative weathering of reduced crustal materials, a process that requires higher O₂ than earlier sinks like volcanic gas oxidation). Alternatively, here, O₂ levels rose until the burial efficiency increased via POOM stabilizing O₂ production at a new, higher level that presumably was then matched by a higher rate of consumption (all requiring higher O₂ levels).

Authors' Response:

We think there exist (at least) two different views of the dynamics of Earth's oxygenation events in the community. The first view [e.g., references (4), (5) and (6) cited in the manuscript] is that Earth's oxygenation is a shift of the biogeochemical steady state to a different level. The second view [e.g., references (8), (9) and (10) cited in the manuscript] is that Earth's oxygenation is a switch between stable states through a dynamic bifurcation, which is also called as "bistability" or "multistability" in the literature [e.g., reference (8) cited in the manuscript]. The difference between these two views is subtle; we clarify this in the following paragraphs.

The first view interprets Earth's oxygenation as a shift in the equilibrium of the global redox state; this is what the "planetary-scale Le Chatelier's principle" (Line 26 in the old manuscript; Line 29 in the revised manuscript) refers to. Following the conventional expression of the net flux of O₂ in

Earth's atmosphere, we can write the rate of the change of O_2 levels as $d[O_2]/dt = F_{source} - F_{sink}$, in which $[O_2]$ is the level of atmospheric O_2 , t is time, d represents the first-order derivative, F_{source} is the source flux of O_2 , and F_{sink} is the sink flux of O_2 [refer to reference (3) cited in the manuscript]. The first view has an implicit assumption of stability and assumes that the equilibrium evolved according to a planetary-scale Le Chatelier principle. Under this view, the atmospheric O_2 levels would have infinitely many steady states because there are infinitely many values of F_{source} and F_{sink} that satisfy $F_{source} = F_{sink}$. This is exactly as what you said in your comments: “*the evolution of its atmospheric level has been through a series of quasi-steady states.*” In other words, under the first view, we completely agree with what you described in your comments.

However, in our study, we took the second view, which assumes the bistability of Earth's atmospheric O_2 levels (i.e., one low-level stable state before the oxygenation and one high-level stable state after the oxygenation) and interprets Earth's oxygenation as a switch of alternative stable states [e.g., references (8), (9) and (10) cited in the manuscript]. Geochemical records suggest that the two major oxygenation events in Earth's history (i.e., the Great Oxidation Event and the Late-Neoproterozoic Oxidation Event) were stepwise following a period of extended stasis. From the perspective of nonlinear dynamics, such an abrupt change in Earth's oxygen cycle suggests a dynamic bifurcation from one low-level stable state to a new, high-level stable state, rather than a shift of the same biogeochemical steady state to a different level.

From your above comments, we indeed have recognized that using “steady state” may cause confusion and misunderstanding. To address your concern, we have changed “steady state” to “stable state” in the whole manuscript, mentioned that our work is based on a view of “multistability (or bistability)”, and changed the following sentence

“However, an alternative possibility exists: oxygenation occurs when O_2 levels switch from one steady state to another.” (Line 26 – Line 28 in the old manuscript.)

to

“However, an alternative possibility exists: the global redox state exhibits multiple equilibria, and oxygenation occurs when O_2 levels dynamically switch from one stable state to another.” (Line 29 – Line 32 in the revised manuscript.)

Also, in our old manuscript (Line 24 – Line 25), we said, “ O_2 levels rise when production exceeds consumption”. From our discussion above, it should be clear that we did not mean Earth's oxygenation would have occurred immediately once its production exceeds its consumption. You can find similar statement in other references; for example, the last sentence in the first paragraph of Goldblatt, *et al.*, *Nature* 443 (2006): 683-686: “... *the Great Oxidation Event was triggered when the oxygen source exceeded the input of volcanic and metamorphic reductants.*” To address your concern and avoid readers' confusion, we rewrote this sentence to “ O_2 accumulates when its production rate exceeds its consumption rate.” (Line 24 – Line 25 in the revised manuscript.)

Line 27: so yes, a switch from one steady state (essentially zero) to a new one. I don't agree though that this requires positive feedback; a temporal decline in the potential consumption rate (e.g., a decline in the volcanic sink) could eventually reduce the sink below the source, allowing O_2 to rise until a new sink was established (as above, e.g., through oxidative weathering).

Authors' Response:

As discussed above, we think there are (at least) two different ways of viewing the dynamics of Earth's oxygenation events; whether positive feedback is required depends on the specific view one takes.

Under the first view (i.e., Earth's oxygenation is a shift of the biogeochemical steady state to a different level), Earth's atmospheric oxygen increases through a series of quasi-steady states and it does not require positive feedback. The second view instead interprets Earth's oxygenation as a switch between stable states through a dynamical bifurcation (i.e., "bistability"). Dynamical bifurcation (or bistability) requires the existence of positive feedback. In the manuscript, we cited a few references [i.e., (8), (9), (10); Line 29 in the old version; Line 33 in the revised version] taking this view (i.e., a switch of stable states of O₂ levels through dynamic bifurcation during Earth's oxygenation). All these references suggested some types of positive feedbacks responsible for the abrupt switch of stable states. For example, reference (8) proposed that, during the Great Oxidation Event, an increase in O₂ levels promoted the formation of ozone layer, which could shield the ultraviolet and decreased the oxidation rate of methane by O₂, leading more O₂ to accumulate in Earth's atmosphere; reference (9) suggested that, during the Late-Neoproterozoic Oxygenation Event, an increase in Earth's atmospheric O₂ levels could promote the bioavailability of phosphorous, which enhanced the primary production of organic matter and therefore the amount of buried organic matter, leading to the further accumulation of O₂ in Earth's atmosphere. Overall, these positive feedback mechanisms (i.e., an initial increase of O₂ levels leads to further increase of O₂ levels) are the bases for the switch of O₂ stable states presented in these studies. Although these purely geochemical positive feedback mechanisms are possible, they may be less responsive to environmental changes than the geological positive feedback intertwined with biological evolution – as we stated in the manuscript (Line 28 – Line 30 in the old version; Line 32 – Line 34 in the revised version). Therefore, in our study, which views Earth's oxygenation as a switch between stable states through a dynamic bifurcation, we suggested a new positive feedback mechanism that is derived from the interactions between life and the environment (i.e., the POOM hypothesis).

In summary, whether positive feedback is required depends on how we view the dynamics of Earth's oxygenation. The viewpoint of "a shift in the equilibrium of the global redox state" does not require positive feedback, while the viewpoint of "dynamic bifurcation" (or "bistability") requires positive feedback. We do not claim that the second view is superior to the first view. In our study, we just took the second view and therefore stated that the alternative view "*requires the existence of one or more positive feedbacks*" in the manuscript.

Line 54: If g_1 ultimately completely degrades under O₂-free conditions, then doesn't this mean that there would have been no organic matter burial in the Archean?

Authors' Response:

The assumption that g_1 ultimately completely degrades under O₂-free conditions does not indicate that there was no burial of organic matter in the Archean. Please see the below reasonings.

First, our mathematical model [Eqs. (1) and (2)] assumes that the g_1 component eventually completely degrades regardless of the presence of O₂ while the g_2 component decays only when O₂ is present. In the O₂-free Archean environment, the *oxygen exposure time* (t_{ox}) in this model

[i.e., Eqs. (1) and (2)] is always zero. In this case, the degradation depicted by Eq. (2) does not occur, and the total amount of eventually buried organic matter is the initial amount of the g_2 component, ag_0 [i.e., the second initial condition in Eq. (3)].

Second, the other equation system [i.e., Eqs. (4) and (5)] in the manuscript is used to demonstrate the conditions under which the positive feedback we suggested (i.e., the POOM hypothesis) works. This equation system, however, is not applicable to the O_2 -free environments. As introduced in the “Persistence of Partially Oxidized Organic Matter” subsection, some microbial enzymes (such as oxygenases) can insert O atoms from O_2 to organic matter and form reactive oxygen-containing functional groups, which enhances the physical protection of organic matter by minerals, promoting the burial of organic matter. The process of partial oxidation of organic matter occurs in the presence of O_2 . Correspondingly, the term “ $k_{12}g_2$ ” in Eqs. (4) and (5) describes the transformation from the g_1 (unoxidized) component to the g_2 (partially oxidized) component. In an O_2 -free environment, this transformation does not occur (i.e., $k_{12}=0$), and Eqs. (4) and (5) become Eqs. (1) and (2). In this case, again, ag_0 is the amount of buried organic matter (as discussed in the last paragraph).

In summary, the simple model of Eqs. (1) and (2) provides a general scenario of organic matter degradation, and the extended model consisting of Eqs. (4) and (5) is used to specify the conditions under which the proposed positive feedback works. Under an anaerobic condition, Eqs. (4) and (5) become Eqs. (1) and (2) because $k_{12}=0$. Also, in an O_2 -free environment, Eqs. (1) and (2) indicate that all g_2 is ultimately buried while all g_1 eventually degrades. Therefore, the assumption that g_1 ultimately completely degrades under O_2 -free conditions does not indicate that there was no burial of organic matter in the Archean.

Line 57: The whole argument of this manuscript is built on the tacit assumption that nutrient supply (e.g., of phosphate) doesn't limit the burial rate of organic matter. That should be stated explicitly.

Authors' Response:

This is a great suggestion. Indeed, many previous studies have suggested that the increase in nutrient supply would promote primary productivity and therefore the burial of organic matter, enhancing the accumulation of O_2 . However, some other work [e.g., Kipp, *et al.*, *Global Biogeochemical Cycles* (2021) 35. e2020GB006707] has challenged this view and suggested that the dramatic burial of organic matter on the ancient Earth was due to the increase in burial efficiency rather than primary productivity (or nutrient supply). This view (“via burial efficiency”) supports the theory presented in our manuscript. To explicitly state this, we have added three new sentences in the last paragraph in the “Positive Feedback in the Ancient O_2 -limiting Sediments” subsection (Line 124 to Line 129 in the revised manuscript) to address this point (i.e., nutrient supply doesn't limit the burial rate of organic matter):

“Enhanced burial of organic matter during Earth's oxygenation events has been attributed to an increase in nutrient supply (e.g., phosphate) that promoted primary productivity (9, 10). The positive feedback mechanism described in this work instead implies that the elevation of O_2 level derived from an increase in organic burial efficiency. This is supported by a recent study suggesting that the increase in burial efficiency rather than primary productivity was responsible for the substantial burial of organic matter in Earth's ancient O_2 -limiting environment (31).”

Line 68: somewhere the authors need to address the claim by Kennedy et al. (2006; Science 311(5766):1446-9) that the types of clay minerals the authors invoke here to be strongly adsorbing POOM didn't appear in the environment until nearly 2 billion years after the Great Oxidation Event, and thus weren't available to support the hypothesized mechanism presented here. I think Kennedy et al. fundamentally misinterpreted the observed trends in clay mineral abundance through time as a primary rather than a diagenetic / metamorphic trend, but to my knowledge no one has ever published a challenge to that paper. I think the current authors need to do so.

Authors' Response:

We agree that clay minerals had not appeared at the time of the Great Oxidation Event according to this paper by Kennedy et al (2006) and some other studies. In the “Persistence of Partially Oxidized Organic Matter” section (Line 76 in the old manuscript; Line 94 in the revised manuscript), we mention clay minerals (and iron oxides) because clay minerals are one type of the common minerals contributing to the physical protection of organic matter in the modern environments, but we did not claim that clay minerals contribute to the burial of organic matter during the Great Oxidation Event. Instead, we only discussed the role of iron oxides in the physical protection of organic matter in the Great Oxidation Event; in the Discussion section (Line 185 – Line 189 in the old manuscript; Line 230 – Line 235 in the revised manuscript), we said: *“Furthermore, with the rise of atmospheric O₂, iron(II) dissolved in seawater or contained in minerals such as pyrite (FeS) (43) was oxidized to iron(III) (44), potentially promoting the physical protection of organic matter (18, 20). As a result, POOM produced by the oxidative metabolisms would have been strongly protected by the newly accumulated iron(III) minerals, thereby enhancing the accumulation of atmospheric O₂.”*

Moreover, we indeed mentioned in the manuscript that clay minerals appeared on Earth's surface in Late Precambrian/Early Phanerozoic – that is, nearly 2 billion years after the Great Oxidation Event. In the Discussion section (Line 202 – Line 204 in the old manuscript; Line 237 – Line 240 in the revised manuscript), we said: *“Oxidative metabolisms likely further spread during the Phanerozoic, promoting POOM formation and O₂ accumulation. This connection may have been facilitated by the biominerals (44) and clay minerals (44, 45) that had accumulated on Earth's surface since the Early Phanerozoic.”* However, according to your suggestion, we cited this paper by Kennedy et al. [i.e., reference (45)] in the revised manuscript.

Moreover, the theory presented in our manuscript is independent of the interpretations in Kennedy et al. [*Science*, (2006) 311(5766):1446-1449]. In our theory, the positive feedback (responsible for Earth's oxygenation) derives from the production of partially oxidized organic matter; this positive feedback can be amplified when the abundance of the minerals contributing to the strong physical protection of organic matter (e.g., iron oxides and clay minerals) increases – as described in the second and third paragraphs in the “Discussion” section in the manuscript. Particularly, our theory suggests that the amplification of the positive feedback in the Early Phanerozoic was caused by the increase in the *abundance* of clay minerals; this does not depend on the specific trends (i.e., a primary trend vs. a diagenetic/metamorphic trend) of clay minerals. However, to address your concern here, we added the following sentence in the revised manuscript (Line 240 -- Line 242):

“Such facilitation should have been caused by the increase in the abundance of clay minerals but does not depend on their specific trends (i.e., whether a primary trend or a diagenetic/metamorphic trend).”

Line 71: POOM is partially oxidized by definition. That means, per unit C buried, it is less effective at net O₂ production. I think this needs to be considered / quantified by the authors. Without the enhanced burial efficiency of POOM argued here, the shift from the burial of non-oxidized organic matter in the Archean to POOM in the Proterozoic would have REDUCED oxygen production! Here again, consideration of C/P ratios along with the oxidation state of the C being buried would be helpful.

Authors' Response:

If non-oxidized organic matter and POOM were buried at the same efficiency (i.e., under your assumption of “*without the enhanced burial efficiency of POOM*”), then yes, partial oxidation would result in a net decrease in net O₂ production because some O₂ is consumed in the production of POOM. However, the POOM hypothesis attributes enhanced burial of organic carbon in Earth's ancient O₂-limiting environments to an increase in burial efficiency rather than an elevation of primary productivity and suggests that burial efficiency is enhanced in response to partial oxidation to a degree sufficient to offset this initial investment of O₂. This is because the "additional" oxygen (i.e., oxygen atoms inserted to organic carbon during the process of partial oxidation) sequestered with POOM occurs only where organic compounds (e.g., biopolymers) are clipped by oxidative enzymes, whereas the number of carbon atoms between such end points is likely much greater than one. As a result, POOM is more effective at net production.

Moreover, we think that considering phosphate (i.e., the “C/P ratios”) is unlikely to provide us more insights. Again, as mentioned in your above comments on line 57 and also in our responses to it, in the context of our theory, it is burial efficiency rather than primary productivity that influences the atmospheric O₂ levels in Earth's ancient O₂-limit environments. In other words, our theory is independent of primary productivity and therefore is not related to P.

Line 89: remove "be" (and by the way, on line 6 and elsewhere, ...ly adverbs don't take hyphens)

Authors' Response:

We have removed “be” in the sentence (Line 116 in the revised manuscript). The hyphens in all “...ly-” adverbs appeared in both main text and supplementary information have been deleted.

Line 94: this statement of the positive feedback could be reversed to say "a decrease in oxygen level can give rise to further decreases via" the same mechanism. Yet the oxidation of the atmosphere apparently was irreversible. I think the authors need to address the reversibility of their positive feedback vs. the irreversibility of atmospheric oxygenation.

Authors' Response:

Our paper does not provide a detailed dynamical model of oxygenation. However, in general, there are two scenarios relative to the question of “reversibility”. (1) The first scenario is that a single stable state shifts as the rates of production and consumption of O₂ change, which is the “global-scale Le Chatelier's principle” in the “Introduction” section refers to. We would call such changes reversible. (2) The second scenario is that there exist multiple stable states (also called as “multistability”); moving from one stable state to another new stable state requires changing one

or more control parameters in the system, (e.g., one such parameter could be the rate at which reducing gases are injected into the atmosphere and oceans). If the changes in the control parameters are monotonic (e.g., if a parameter is always decreasing or never increases), then the shift from one stable state to another could be effectively irreversible. However, if the control parameter can, say, decrease after it increases, then it is possible for the system to restore its original equilibrium. We are preparing another paper that presents a dynamical system of Earth's oxygen and carbon cycles in the past billions of years, which quantitatively discusses the switch between alternative stable states of Earth's atmospheric O₂ levels as some control parameters change.

Moreover, the positive feedback we suggested only operates in environments where the O₂ levels remained low and the O₂ exposure time is short (as depicted by the blue dashed line in Figure 2B in the manuscript). After the atmospheric O₂ levels reach a high-level *stable* state, it will be regulated by a negative feedback mechanism (as depicted by the blue solid line in Figure 2B in the manuscript), which would stabilize the atmospheric O₂ at a high-level state. As a result, a decrease in O₂ level would not lead the new, high-level stable state would not return to the old, low-level stable state. To address this point, we added the following sentences in the "Positive Feedback in the Ancient O₂-limiting Sediments" subsection in the revised manuscript (Line 118 - Line 123):

"The positive feedback would eventually be taken over by a negative feedback (blue solid curve in Fig.2B) when $t_{ox} > t_{ox}^$ (i.e., when the atmospheric O₂ reaches higher levels and the sedimentary environments become more oxygenated) because nearly all organic matter, protected by minerals or not, eventually degrades after long-term exposure to O₂. This negative feedback would stabilize O₂ at a new, high-level stable state after Earth's oxygenation."*

(I'm afraid I have no expertise to comment on the phylogenetic analyses)

Line 140: "excursion"

Authors' Response:

We have corrected the typo -- "*excursion*" – to "*excursion*" (Line 185 in the revised manuscript).

Lee Kump, Penn State

Reviewer #2 (Remarks to the Author):

The manuscript on Oxidative metabolisms catalyzed Earth's oxygenation by Shang, Rothman and Fournier deals with an important part of the evolutionary history known as Great Oxidation Event. Although the presented hypothesis is quite interesting it needs more support and further complex & comprehensive considerations in comparison with published alternative explanations of this evolutionary event on Earth.

From these 16 points it is clear that I suggest MAJOR REVISION of this manuscript to be considered and I can review it again after improvements.

Authors' Response:

Thank you very much for your thoughtful and constructive comments.

These are important points that authors need to consider for preparation of an improved version of this manuscript:

1. Immediately at beginning of the abstract, lines 3-4: "GOE is generally explained by the burial of reduced organic matter..." - but it is primarily explained by different isotopic signatures in rocks older and younger than 2.4 billion years suggesting a significant difference in the content of Earth's atmosphere (see e.g. *Nature* Vol. 443 (2006) pp. 643-645 and references therein.). Only secondarily the proposed organic-carbon burial would trigger GOE. This needs to be explained more clearly for those readers that are not deeply involved in this topic.

Authors' Response:

We think this criticism is partially due to our different understanding of what an "explanation" of Earth's oxygenation is. We agree that "*different isotopic signatures in rocks older and younger than 2.4 billion years suggesting a significant difference in the content of Earth's atmosphere*". These changes in different isotopic signals suggest (1) the *timing* of Earth's oxygenation events occurred (e.g., the Great Oxidation Event occurred around 2.4 billion years ago) and (2) *how much* the levels of the atmospheric O₂ had changed during the oxygenation events (e.g., O₂ increased from around 10⁻⁶ PAL to around 10⁻² PAL during the Great Oxidation Event). However, we think they do not explain *how* and *why* the oxygenation events had occurred. Instead, Earth's oxygenation is usually explained by the decline of the reducing agents in Earth's surface environments (the common source of reductants is volcanic gases) or an increase in the source of O₂, (such as the dramatic burial of organic matter). As the paper that you mentioned in your comments [i.e., Kasting, *Nature* 443 (2006): 643-645] said, Earth's oxygenation is usually explained by a decrease in reductant input (e.g., reducing gases from volcanoes) or an increase in O₂ input (e.g., the burial of organic carbon).

However, we do agree with you that organic-carbon burial is not the only explanation for Earth's oxygenation. And particularly, it is not appropriate to say that the burial of organic matter is a general explanation for the Great Oxidation Event. We have rewritten the following sentence

"The Great Oxidation Event (GOE) is generally explained by the burial of reduced organic matter, which prevents its remineralization via oxygen-consuming processes." (Line 3 – Line 4 in the old manuscript)

To

"The burial of organic carbon, which prevents its remineralization via oxygen-consuming processes, is considered one of the causes of Earth's oxygenation." (Line 3 – Line 4 in the revised manuscript).

In the new sentence, we say "*Earth's oxygenation*" instead of "*the Great Oxidation Event*" because (i) the burial of organic matter carbon has been considered a cause of other oxygenation events (e.g., the Late Neoproterozoic Oxidation Event) as well; and (ii) as we have discussed in the manuscript, the POOM hypothesis is probably also applicable to interpret other oxygenation events (i.e., implied by the temporal correlation between the rapid diversification of BVMOs and the rise of the atmospheric O₂ levels).

Due to the limit of words in the Abstract, we added the following sentence in the first paragraph (Line 25 -- Line 27 in the revised manuscript), which briefly introduced other primary mechanisms of the GOE (e.g., oxidation of Earth's mantle or crust):

“For example, it has been suggested that the GOE might have resulted from the oxidation of Earth's mantle or crust, which reduced the O₂ consumption rate; the dramatic burial of organic matter, which increased the O₂ production rate; or other mechanisms (1-3).”

2. Abstract lines 9-10 authors need to present here clearly which is this "one key enzyme family that generates POOM in marine systems". This is essential for topical searches of potentially interested readers.

Authors' Response:

We agree with this nice suggestion and have rewritten the following sentence

“... we reconstruct the evolutionary history of one key enzyme family that generates POOM...”
(Line 9 – Line 10 in the old manuscript)

to

“...we reconstruct the evolutionary history of one key enzyme family, flavin-dependent Baeyer-Villiger monooxygenases, that generates POOM ...” (Line 9 – Line 11 in the revised manuscript).

3. Line 38 on page 2: "In modern sediments, POOM is produced by aerobic metabolisms with the aid of oxygenases" this is claimed without any citation. So are oxygenases just to aid this process or are they indeed key enzymes that allow production of POOM?? Could potentially other enzymes besides oxygenases also be involved in this process?

Authors' Response:

We agree that a more recent reference should be provided after this sentence. I have added one in the revised manuscript [i.e., reference (12)], which was published in year 2021.

The word “aid” in this sentence means “catalysis”; it does not indicate that oxygenases are not important in the formation of POOM. As we discussed in the “Persistence of Partially Oxidized Organic Matter” subsection in the manuscript, oxygenases play a key role in the production of POOM. But, from your comments, we indeed have recognized that this word (i.e., “aid”) may lead confusion and misunderstanding.

Regarding your concern about “other enzymes”, we agree that the formation of POOM requires a series of reactions catalyzed by different enzymes, although the oxygenases are a key family for POOM production. As we said in the manuscript, *“one group of enzymes that can catalyze the formation of intermediate products in oxidative metabolisms are oxygenases.”* (Line 99 – Line 101 in the old version; Line 137 -- Line 139 in the revised version.) Also, we agree that more broad scope and statements should be provided in the Introduction section and the specific instances should remain in the Results and Discussion Section. So, we have replaced “oxygenases” by “oxidative enzymes”, which include border families of enzymes that catalyze the oxidative degradation of organic matter.

Overall, to address your concerns above, we have rewritten the following sentence

“In modern sediments, POOM is produced by aerobic metabolisms with the aid of oxygenases.”
(Line 38 in the old manuscript)

to

“In modern sediments, POOM is produced by aerobic metabolisms catalyzed by oxidative enzymes (12); here we focus on a representative, POOM-producing enzyme – flavin-dependent Baeyer-Villiger monooxygenase (BVMO) (13).” (Line 42 – Line 44 in the revised manuscript).

4. Lines 40-41: authors selected Baeyer-Villiger monooxygenase for further work. It needs to be added here that they are flavin-dependent enzymes. But why not mention also some alternatives e.g. P450 monooxygenases that are heme-dependent and are also expected to have very ancient origin. Could alternative oxygenases not contribute to this important process just "BVMO"?

Authors' Response:

We agree that we should making it clear to the readers that Baeyer-Villiger monooxygenases are flavin-dependent enzymes. To address this point, we have added “flavin-dependent” here (i.e., Line 42 - Line 44 in the revised manuscript): “... *here we focus on a representative, POOM-producing enzyme – flavin-dependent Baeyer-Villiger monooxygenase (BVMO) (13).*”

Regarding the heme-dependent P450 monooxygenases, we think their evolutionary history should not be related to the POOM hypothesis although they are expected to have very ancient origin. Heme-dependent P450 monooxygenases are almost exclusively used in anabolic processes in microbes (biosynthesis of various lipids and related compounds) [e.g., Greule, *et al. Natural product reports*, 35(2018): 757-791]. In other words, they are not used in microbial energy metabolisms; the intermediate products produced by the anabolic metabolisms catalyzed by the heme-dependent P450s are converted to other products, intracellularly (e.g., carotenoid biosynthesis). So, their activities are not expected to generate POOM, which is eventually buried in sediments. In contrast, BVMOs have been shown to be used in catabolic processes in microbes [e.g., Tolmie, *et al. Natural product reports*, 36(2019): 326–353] and to be particular in the function of catalyzing the formation of POOM that persist in marine environment [reference (13) cited in our manuscript]. This is the reason why BVMOs are the gene family of choice for using as a proxy in tracing POOM in our study.

Moreover, the paper by Landry *et al.* [i.e., reference (13) cited in our manuscript] simply pointed out that, generally, P450 monooxygenases are involved in forming oxygen groups on alkanes, which is not necessarily catabolism, and an important part of many lipid biosynthesis pathways, as extensively detailed in the review by Greule, *et al.* [*Natural product reports*, 35(2018): 757-791]. Landry *et al.* referenced sterol degradation as a possible catabolic pathway, which is the only referenced evidence of P450 monooxygenases being involved in degrading recalcitrant organic matter. Also, this referenced example (e.g., sterol degradation) is from a single study in which a soil bacterium from Actinobacteria was shown to degrade sterols using P450 monooxygenases, which is apparently common in other soil Actinobacteria as well. However, our study (and also the POOM hypothesis) focus on the degradation and preservation of organic matter in marine

sediments because most of the organic matter burial on Earth occurs in marine sediments [e.g., Galvez *et al.*, *Nature Geoscience*. 13, 535–546 (2020)]. Therefore, the degradation of sterol in soils should not be relevant to the POOM hypothesis suggested in our manuscript.

Overall, the above reasons suggest that the heme-dependent P450 monooxygenases should not be expected as a potential additional record for POOM although they have ancient origin. There might be some other oxygenases that is involved in the formation of POOM, but we focus on the BVMOs here. This is not only because that BVMOs are able to catalyze the formation of POOM, but also because that the previous studies (as mentioned above) have shown that the POOM-producing oxidative metabolisms catalyzed by BVMOs are catabolic and occur in marine systems.

5. Line 52 on page 3: "labile component g₁ and recalcitrant component g₂" - this seems a very simplified scheme but the most important point of criticism here is that - most organic components do not decay in the presence of molecular O₂ easily. This mathematical model therefore appears to be a comfortable simplification with respect to expected real conditions in primordial Earth surface. Molecular oxygen is not so reactive itself but there are the so called reactive oxygen species ("ROS") i.e. hydrogen peroxide, superoxide radical, hydroxyl radical, hydroperoxyl radical, nitric oxide radical and last but not least singlet oxygen - all products of metabolism, or present in the environment that react with organic components readily.

Authors' Response:

We agree that many organic compounds cannot be easily oxidized by molecular oxygen *directly*. But their oxidation by molecular oxygen would become (much) easier in the presence of microbial enzymes, which could reduce the activation of degradation. As you see, microbial enzymes (in our manuscript, oxygenases) are a key component in the POOM hypothesis we proposed and are also frequently mentioned when we discuss organic matter degradation in the manuscript. And more important, although the concept of *oxygen exposure time* – that is, t_{ox} in our mathematical model (Line 53 in the old manuscript; Line 68 in the old manuscript) – has been considered a proxy of organic matter degradation [refer to references (15), (16) and (20) cited in the manuscript], it does not indicate that all organic matter easily decays in the presence of O₂. Our mathematical model assumes that a portion of organic matter (the recalcitrant component g₂) can only decay aerobically [i.e., Eq. (2) in the manuscript]; that is, our model *only* assumes that the presence of O₂ is a *necessary* condition for *a portion* of organic matter (i.e., the recalcitrant portion g₂) to decay, but does not claim that the presence of O₂ is a *sufficient* condition for the degradation of the recalcitrant component.

Moreover, the reactive oxygen species were unlikely to have obvious influence on the degradation and preservation of organic matter in the ancient O₂-limiting environment. Molecular oxygen (O₂) is a major source of reactive oxygen species (Ma, *et al.*, *ACS Earth Space Chem.*, 2019, 3: 73 - 747; Waggoner, *et al.*, *Geochimica et Cosmochimica Acta*, 2017, 208: 171-184). In the ancient O₂-limiting environment, it is likely that reactive oxygen species would not preferentially form and therefore would have remained at low levels. In Earth's ancient reducing environment, reactive oxygen species would preferentially react with those abundant, more strongly reducing small molecules instead of POOM.

6. Line 60 on page 4: when writing about conundrum, it is even much more complicated if we consider point 5. above. references #15 & #16 are quite old and partially speculative. Besides the effect of POOM some other factors derived e.g. from the presence of reactive oxygen species can have had influence on particular (reactive) oxygen concentrations that were probably not identical and homogenous over the whole Earth's surface.

Authors' Response:

Indeed, these two references were published more than twenty years, but they are classical and standard literature in the field of "organic matter preservation in sediments" (this is obviously reflected by their high citation numbers). More important, these references are the earliest literature proposing the oxygen exposure time hypothesis, which suggested that (1) a portion of organic matter only decays in the presence of O₂ (this inspires our mathematical model in the manuscript), and (2) negative feedback stabilizing the modern atmospheric O₂ levels. Many viewpoints in these two papers, including the oxygen exposure time hypothesis, have been supported by many studies in the past two decades [please refer to reference (18) (a review) cited in our manuscript]. To address your concern, we have added reference (18) here (Line 78 in the revised manuscript).

Regarding the possible influence of the reactive oxygen species on the O₂ levels, we think such influence is not obvious. In addition to the reason that we have discussed in the responses to your comment # 6 (i.e., a major source of reactive oxygen species – that is, O₂ – remained at very low levels in the ancient O₂-limiting environment), there are other three reasons for this:

(1) Reactive oxygen species are short-lived molecules with half-lives on the order of seconds to days (Hansel and Diaz, *Annual Review of Marine Science*, 2021. 13: 177–200), which is much shorter compared to the length of time for (recalcitrant) organic matter to decay. The short half-lives of the reactive oxygen species indicate that they are unlikely to accumulate to significant amounts in (aqueous) environments and instead they are likely to react only in the location where they are formed. The transient reactive oxygen species would eventually accumulate as much more stable O₂. These imply that reactive oxygen species are unlikely to have apparent influence on the global-level preservation of POOM.

(2) Reactive oxygen species can be generated inside the cells, but they are rapidly removed by catalases and peroxidases within cells, or by reacting with other materials (e.g., causing damage to organic biomolecules) (Fridovich, *The Journal of Experimental Biology*, 1998, 201: 1203–1209). These would limit the amount of the intracellular reactive oxygen species.

(3) Reactive oxygen species can also be generated outside the cells (Hansel and Diaz, *Annual Review of Marine Science*, 2021. 13: 177–200). The oxidation of (recalcitrant) organic matter by these (extracellular) reactive oxygen species (e.g., hydrogen peroxide) requires enzymes (e.g., peroxidases) (Burdige, *Chem. Rev.* 2007, 107: 467–485) as well. The recalcitrance of organic matter is generally attributed to the intrinsic components (and structures) and the external (i.e., physical) protection (Arndt, *et al.*, *Earth-Science Reviews*, 2013. 123: 53-86). Indeed, reactive oxygen species are stronger oxidants compared to O₂ and are more effective at oxidizing organic matter that is intrinsically recalcitrant. However, the POOM hypothesis suggests that the partial oxidation enhances the external protection of organic matter because the partially oxidized organic matter is more constrained in space and has less accessibility to microbial enzymes compared to the unoxidized organic matter. Therefore, in the context of the POOM hypothesis, the strong physical protection by minerals in sediments should effectively protect POOM from being oxidized by the reactive oxygen species (e.g., hydrogen peroxides) via reducing the accessibility

of the enzymes (e.g., peroxidases). Moreover, if reactive oxygen species such as peroxides were really produced in some (small) quantity in the ancient O₂-limiting environments, they might indeed be an alternative pathway to POOM production, mediated by enzymes or not; the relevant process still holds: O₂-limiting environments result in POOM and therefore an increased burial of organic carbon. This process of POOM production could have been occurring alongside biologically mediated partial oxidation in the ancient O₂-limiting environments.

7. line 66: this is a very general statement "secrete enzymes to degrade organic matter". Namely, some enzymes like hydrolases do not need to use oxygen to make their efficient reaction of organic matter degradation.

Authors' Response:

Yes, indeed we agree that "*secrete enzymes to degrade organic matter*" is a general statement and also agree that "*some enzymes like hydrolases do not need to use oxygen to make their efficient reaction of organic matter degradation*". We made this general statement because the goal of the three sentences at the beginning of this paragraph (i.e., Line 65 – Line 70 in the old manuscript; Line 83 – Line 88 in the revised manuscript) is to provide a general overview of organic matter degradation and preservation, rather than introducing the POOM hypothesis or anything related to oxygenases and O₂. Therefore, we think there is no need to make a specific statement related to oxidative enzymes at this place.

8. Figure 1 on page 5 - if really conceptual this figure certainly needs an upgrade. What kind of enzymes are considerable for this scheme? Divergent oxygen-containing functional groups need to be clearly labelled and distinguished here otherwise this figure looks just like a sketch from an undergraduate-level book.

Authors' Response:

We have revised both Figure 1 and its caption to address your concerns and to provide more detailed description/explanation for the POOM hypothesis (i.e., how the partial oxidation of organic matter impedes its accessibility to microbial enzymes and enhances its potential for long-term preservation). In the revised version, we have labelled carbon-degrading enzymes and the representative reactive oxygen-containing functionals (i.e., carboxyl and hydroxyl groups) that are produced in partial oxidation and contribute to strong association of organic matter with minerals. In addition, in the new version of Figure 1, we have distinguished the "exposed enzyme targets" and "protected enzyme targets" to demonstrate the changes of the biopolymer due to partial oxidation, and have also added more enzymes (i.e., blue pieces) on the left panel to more effectively show that the enzyme targets are more accessible to microbial enzymes before partial oxidation.

The caption of Figure 1 has been rewritten as below:

"Comparison of biopolymers and their interaction with mineral surfaces before (A) and after (B) partial degradation by oxidative metabolisms. In (A), only one site (yellow oval) of the biopolymer is sorbed to the mineral surface (horizontal line) and the exposed enzyme targets on the biopolymer

are freely accessible to carbon-degrading enzymes secreted by microorganisms. In (B), reactive oxygen-containing functional groups such as carboxyl groups and hydroxyl groups formed by partial oxidation in the presence of O₂ create additional sorption sites (e.g., R-COO⁻ or R-OH) that enhance the association of the shorter organic carbon chains with the mineral surface. These partially oxidized, shorter organic carbon chains in (B) are more constrained compared to (A); consequently a large portion of enzyme targets on these shorter organic carbon chains are relatively inaccessible to microbial enzymes. Compared to (A), their degradation requires more investment of free energy to overcome the energy barrier that prevents enzyme access. The juxtaposition of (A) and (B) shows how partial oxidation impedes the biopolymer's accessibility to microbial enzymes and enhances its potential for long-term preservation.”

9. Line 101 - Reference #30 from the year 1992 is really very old. Was there no published progress in the meantime? Most oxygenases are either flavin or heme dependent. Was the availability of these essential cofactors before and during GOE for the proposed POOM formation guaranteed as well? This could be followed with the parallel phylogeny of flavin and heme producing pathways (or at least discussed shortly).

Authors' Response:

To address your concern about the old (i.e., year 1992) reference [i.e., reference (30) in the old manuscript], we have replaced it by reference (31) in the revised manuscript; this reference is a review of oxygenases published in year 2018.

Regarding your concern about the age of flavin cofactors, some studies have shown that some redox-active flavin-containing coenzymes, such as the FAD and FMN cofactors for BVMOs, likely have very ancient origin [e.g., Caetano-Anolles, *et al. Journal of Molecular Evolution* (2012) 74:1–34; Wang, *et al. Molecular Biology and Evolution*, 28 (1): 567–582]. The molecular clock-based analysis by Wang, *et al. [Molecular Biology and Evolution*, 28 (1): 567–582] has suggested that these flavin-containing cofactors had already existed before the advent of the Great Oxidation Event. Moreover, some work even hypothesized that FAD and FMN (cofactors for BVMOs) to be part of the prebiotic "RNA World" and subsequently passed down to proteins because they are nucleotide-derived [Cochrane and Strobel, *RNA* (2008) 14: 993-1002]. To address your concern, we have added the following sentences in the revised manuscript (Line 154 -- Line 157):

“Previous studies have suggested that some flavin-containing cofactors, including flavin adenine dinucleotide and flavin mononucleotide that are utilized by BVMOs, had already existed before the advent of the GOE (38) or even as old as the age of the "RNA World" (39). However, the evolution of BVMOs has rarely been explored.”

10. Line 125 and Supplementary Figures S1-S4. The fonts used in supplementary figures are so small that they are really hardy readable even under a huge magnification. In some cases they are even overlapping. Most readers will be impatient to study for hours the important details of these evolutionary figures. Thus the claimed HGT events are not really obvious from the current presentation of these results (e.g. HGT event #1 described in line 134). Authors need to label clearly the directions of HGT between various bacterial phyla. Moreover, Table S5 just presents

some numbered nodes for respective donors and recipients. This is almost impossible to follow and resume for an average reader. It shall be clearly declared - beyond the numbered nodes in this table - if a particular HGT occurred e.g. from Cyanobacteria to Chloroflexi etc.

Authors' Response:

These are great points! We have improved our manuscript and SI accordingly.

We have labelled the phyla in the species trees – that is, Figures S1, S3 and S5 in the revised Supplementary Information. A new figure illustrating the directions of the oldest HGT events (i.e., HGT #1 and #2, whose older age bounds are older than 2000 Ma) is provided in Supplementary Information (i.e., Figure S7). We did not show all 68 inferred HGT events in Figure S7 because doing so would make the lines indicating HGT directions cross one another and hard to tell the HGT directions. Instead, we provided a new table – that is, Table S6 in the revised Supplementary Information – to demonstrate the phyla of the donor and recipient of each HGT event (as you suggested in the above comments).

Correspondingly, we have revised the following sentences in the in “Phylogenetic Analyses” in the manuscript (Line 170 – Line 177 in the revised manuscript):

“To investigate the relevance of these HGT events to the evolution of oxygen and carbon cycles, we construct the weighted distributions of the older and younger bounds for the timing of 68 HGT acquisitions of the BVMO gene that have bootstrap values $\geq 80\%$ (Materials and Methods section). ... The age information of these HGT events is presented in Supplementary Materials, Table S5 and graphically summarized in Fig. 3. Table S6 (in Supplementary Materials) presents the directions (i.e., donors and recipients) of these HGT events, and Figure S7 (in Supplementary Materials) graphically illustrates the directions of some representative (i.e., the oldest) HGT events.”

Besides, we have added the following sentences in the caption of Figure 3 in the revised manuscript:

“... The directions (i.e., donors and recipients) of these HGT events are provided in Supplementary Materials, Table S6. ... The initial HGT (also illustrated in Supplementary Materials, Figure S7) acquisition occurred on the branch between stem SAR202 node (red filled circle) and crown node SAR202 (blue filled circle). ...”

11. Figure 4 on page 11 - the authors mention also a rapid divergence of Eukaryotes that reveals also some differences in timing in various literature sources. Can the evolution of Eukaryotes and their smarter metabolism have also an significant influence & impact on the atmospheric oxygen level and the production of POOM?

Authors' Response:

We agree that there is no consensus on the timing of the divergence of Eukaryotes. However, as we have mentioned in the “Phylogenetic Analyses” section (i.e., Line 159 in the old manuscript; Line 204 in the revised manuscript) and also the “Discussion” section (e.g., Line 199 in the old manuscript; Line 247 in the revised manuscript), we are referring to the diversification of eukaryotic marine algae (as indicated by molecular clock, microfossil, and lipid biomarker records; e.g., refer to Sánchez-Baracaldo *et al.*, *PNAS*, 114 (2017): E7737-E7745) rather than general Eukaryotes. We apologize that we did not clearly state this (i.e., “eukaryotic marine algae”) in the

caption of Figure 4 in the old manuscript. To avoid confusion and misunderstanding, we have changed the following sentence in the caption of Figure 4:

“... the time of the GOE, the rapid divergence of Eukaryotes, and the Permo-Carboniferous O₂ pulse.”

to

“... the time of the GOE, the rapid divergence of eukaryotic marine algae, and the Permo-Carboniferous O₂ pulse.”

Moreover, as discussed in the manuscript (Line 198 – Line 206 in the old manuscript; Line 246 – Line 255 in the revised manuscript), we are speculating that diversification of eukaryotic marine algae could potentially explain the Middle/Late Mesoproterozoic peak shown in Figure 4A, as it is the almost only conspicuous, well-studied major revolution in marine biodiversity during this geologic time interval and would certainly increase the flux of buried organic materials to sediments. Since we do not have direct evidence for this connection, we proposed a possible interpretation for the temporal coincidence of the Middle/Late Mesoproterozoic peak shown in Figure 4A and the timing of eukaryotic marine algae diversification reported in previous studies.

12. Lines 133 onwards - the presentation of observed HGT events is dominantly focused on SAR202 bacteria. But from the first glance of presented evolutionary tree it is obvious that BVMO genes are present also in other (important) bacterial phyla namely: Cyanobacteria, Proteobacteria, Actinobacteria, Bacteroidetes to name just few of them. Could have more divergent bacterial phyla beyond SAR202_Chloroflexi lineage also contributed to the global formation of POOM?? At least some more comprehensive evaluation in this respect is needed here.

Authors' Response:

We apologize for the confusion here. We indeed evaluated the HGT events on the *whole* phylogenetic tree and reported the HGTs that have high bootstrap supports ($\geq 80\%$) in Figure 3 in the main text. In other words, the 68 HGT events shown in Figure 3 are for the whole tree, not just for the SAR202 bacteria only. Now this should be clear from Table S6 in the revised Supplementary Information. Although our analyses indeed detected some HGT events that might have occurred within/between Actinobacteria and Proteobacteria, we did not discuss much about these HGT events in the manuscript because (1) the Actinobacteria in our phylogenies, including *Mycobacteria*, *Streptomyces*, *Mycobacterium*, *Gordonia*, etc., are basically soil bacteria; (2) some portion of the Proteobacteria in our phylogenies (e.g., *Bradyrhizobium*) are soil bacteria, but as we mentioned in the responses to your comment # 4, most of the organic matter burial on Earth occurs in marine sediments; (3) there are some marine Proteobacteria in our phylogenies (more specifically, one species in the Pelagibacterales order and seven species in the Rhodobacterales order), but no previous work has shown that these marine Proteobacteria process recalcitrant organic materials or are able to produce POOM in marine environments (rather, they have been shown to oxidize single carbon compounds (Sun, *et al. PLoS One*, 2011 6(8): e23973)). Therefore, these Actinobacteria and Proteobacteria species are unlikely the major contributors to POOM formation in marine systems and the HGTs within/between them unlikely reflect the diversification

of POOM-producing metabolisms in marine environments. Accordingly, we focus on the SAR202 bacteria for their demonstrated ecological/geochemical importance and relevance to the POOM formation in marine environments, as discussed in the second paragraph in the “Phylogenetic Analyses” section in the manuscript.

Moreover, we have realized that the first sentence in old caption of Figure 3 is not clear. In the revised manuscript, we changed it to

“The weighted means and 95% CIs of the older and younger time bounds for 68 inferred HGT events in the whole phylogeny (main figure), a chronogram of SAR202 lineages within Chloroflexi (Inset A), and the date distributions of the older and younger age bounds for the initial HGT event into SAR202 (Inset B).”

In additions we have added/revised two sentences in the revised manuscript to address that the presented HGT events are not only for SAR202 bacteria.

(1) The following sentence has been added (Line 390 – Line 391 in the revised manuscript):

“These HGT events are within or between the Chloroflexi, Actinobacteria, and Proteobacteria phyla.”

(2) Line 189 – Line 191 in the new manuscript, we revised the old sentence to:

“Fig. 3 also shows that the HGT events of BVMO genes in the whole phylogeny span the Proterozoic and Phanerozoic, apparently increasing in frequency starting in the Late Neoproterozoic.”

13. Lines 179-181 on page 12: "emergence of aerobic metabolic pathways" this is just a very general declaration. It is desirable that authors confront diverse bacterial phyla (mentioned in point 12.) and their proposed ancestors at the estimated time of "Late Archean" to discuss their respective capacity in such aerobic metabolic pathways.

Authors' Response:

As we response to your comment #12, we indeed had evaluated the HGT events on the *whole* phylogenetic tree rather than only the SAR202 clade. To address your concern, we have changed the following sentence:

“In the Late Archean, the initially low and localized O₂ production likely instigated the emergence of aerobic metabolic pathways.” (Line 179 – Line 180 in the old manuscript)

to

“In the Late Archean, initially low and localized O₂ production likely instigated the diversification of aerobic metabolic pathways, such as the POOM-producing oxidative metabolisms. This is supported by the early divergence of BVMO genes in different microbial lineages (Figure S3 and S4, Supplementary Materials).” (Line 223 – Line 226 in the revised manuscript).

14. Lines 185-186 on page 13: Iron II was oxidized to Iron III. In context with ROS mentioned in point 5. - if just traces of peroxide were present around the ancestral POOM producing cells, then the so called Fenton reaction could have occurred with some consequences that would not be as protective as declared here.

Authors' Response:

Previous studies have shown that Fe(III) minerals are able to adsorb more organic matter than Fe(II) minerals, and the association of organic matter with Fe(III) is stronger than Fe(II) minerals [e.g., Barber, *et al. Scientific Reports*, 7, 366 (2017); Nierop, *et al. Science of the Total Environment*, 300 (2002): 201-211.]. This suggests that the transformation of Fe(II) minerals to Fe(III) minerals would enhance the physical protection of organic matter in sediments.

In a Fenton reaction, iron(II) is oxidized by peroxide to Fe(III) [e.g., Barbusinski, *Ecological Chemistry and Engineering. S*, 16 (2009): 347-358]. In other words, a Fenton reaction would promote the formation of iron(III). Therefore, if peroxide were really “present around the ancestral POOM-producing cells”, the Fenton reaction would have enhanced the physical protection of organic matter, promoting the accumulation in Earth's atmosphere.

15. Lines 252-253 on page 16: "they have high predicted structural similarity" - this needs some quantification or more exact description of critical parameters to be able to repeat this procedure on Phyre2 or alternative homology modelling servers. And what were the optimal (optimized) parameters for the alignments? Eventually a typical Phyre model could be presented as supplementary material.

Authors' Response:

We agree with your comments here and have modified the manuscript and SI accordingly.

Regarding the “parameters”, the users of Phyre2 can set only one parameter – that is, the “modelling mode”. To provide more exact description about the Phyre2, we have changed the sentences related to Phyre2 in the manuscript to:

“... we employed the method used in (13) and ran the Phyre2 structural homology recognition server (52) with the ‘normal modelling mode’. It turned out that the query sequences were modelled with 100% confidence and high coverage (>=85%) by the BVMO template in the Phyre2 system, suggesting that those sequences have high predicted structural similarity to BVMOs.” (Line 311 – Line 315 in the revised manuscript.)

Also, according to your suggestion, we have added a Phyre2 model of BVMO in the Supplementary Information; please see Figure S9. And the following sentence has been added to the revised manuscript (Line 315 – Line 316):

“A typical BVMO model generated by the Phyre2 server with the ‘normal modelling mode’ is provided in Supplementary Materials, Figure S9.”

Regarding the sequence alignments, we have changed the following sentence:

“The program MAFFT (50) was used to align the sequences.” (Line 252 – Line 253 in the old manuscript)

to

“The sequences were aligned using the MAFFT program (50) with the progressive method ‘FFT-NS-2’ and ‘--bl 62’ (i.e., ‘the score matrix BLOSUM62).” (Line 316 – Line 318 in the revised manuscript).

16. Line 255 "were concatnated" shall be written as concatenated. This is just a small typing error but the option concatenated sequences could alternatively be used also for multiple oxygenases...

Authors' Response:

Thanks for pointing this out. We have corrected the typo -- *“concatnated”* – to *“concatenated”* (Line 319 in the revised manuscript). We think we should not concatenate oxygenases because this presumes that the oxygenases have a congruent evolutionary history. However, this is obviously not the case; also, the point of HGT detection is that their evolutionary history is not congruent.

Reviewers' Comments:

Reviewer #1:

Remarks to the Author:

I feel that the authors have done a nice job of responding to my initial concerns, with one exception. They've missed the point about phosphate.

Perhaps best articulated by Tyrrell (Nature, 400, 525–531, 1999), P is considered the "ultimate" limiting nutrient, not for productivity, but for organic matter burial. The authors miss this point in their response to my comments arising from line 57 in the original manuscript when they say "Indeed, many previous studies have suggested that the increase in nutrient supply would promote primary productivity and therefore the burial of organic matter, enhancing the accumulation of O₂." The point is that P is buried with organic matter, so the rate of burial can be limited by the supply of P to the ocean (modified by the C/P ratio of buried organic matter, which I bring up later w.r.t. line 71). It's not that increasing P delivery increases primary (or even net production); indeed, these can be independent of P.

So the authors are assuming that P supply does not limit organic matter burial (net oxygen production) and so should state that explicitly. Only if increasing burial efficiency increases the C/P of buried organic matter can it lead to an increase in net oxygen production.

Lee Kump

Reviewer #2:

Remarks to the Author:

The revised manuscript on Oxidative metabolisms catalyzed Earth's oxygenation by Shang, Rothman and Fournier was completely updated and improved according to previous comments of reviewers. However, there are still some aspects that need further clarification. In principle, they can be classified in two divisions: first the aspect of presentation of some results and second the methodological aspect.

To solve the hard conundrum of POOM, as authors describe in the abstract on line 6 the results need to be presented more globally (to my opinion). Yes, authors made already several improvements in this respect but e.g. Figures 3 & 4 are still somehow narrowly focused (maybe this is enough for some specific journal but here a more comprehensive view is expected). Therefore, it is suggested that in Figure 3A not just the ancestor of SAR202 and its descendants are presented. There are really many HGT events demonstrated also with a high statistical support but at least it should be stated whether such HGTs are specific only for SAR202 bacteria or are general property of this gene family for (almost) all marine bacterial phyla? From current presentation it is not really obvious. Figure 4 – for a reader not deeply involved in this topic it is hard to imagine the significance of this output. At least a comparison of this diversification rate of SAR202 BVMO genes with some other (metabolically similar) gene family (presented in other literature?) is needed to get an opinion if such a diversification is comparable with other protein coding families? And there still remains the (main) question whether SAR202 bacteria were the main (or even sole) contributors for a massive POOM burial? There are numerous cyanobacterial marine species. Could have also their ancestors (that should appear rather closely related with Chloroflexi due to several global reconstructions) contributed to POOM burial to some extent? In stromatolites they are present significantly. This does not mean that authors need to extend their phylogenetic analysis for this particular manuscript. Just few sentences about BVMO genes outside the SAR202 group would be appreciated (to follow their origin before multiple HGT events already presented here). In Figure S5 we indeed see „CMS group" that is sister clade with Chloroflexi but unfortunately in Figure S6 that shall refer to BVMO genes it is not as clearly labelled and the font used is really very small...

Minor points about the methodology for consideration:

Lines 303-304: „Protein Basic Local Alignment Search Tool (BLASTp) on the National Center for Biotechnology Information (NCBI) database was used to search the genes of interest.“ But the BLASTp option is dedicated for search of PROTEINS not genes. Also the output is then formed with protein files (sequences). Have the authors searched further really for genes or just for encoded proteins? This needs to be clarified.

Lines 310-314 it is now appreciated that for those oxygenase sequences that were not well annotated in the original database homology models were produced to verify their suitability. But it is not clear and it shall be specified (at least roughly) how many from those 330 (verified) protein sequences do not belong to the SAR202 cluster?

Lines 318-319 „poorly aligned regions were manually deleted“ – this can eventually be an important aspect. If we refer to Phyre2-homology model from Figure S9 where also a pairwise alignment is present – can we at least expect that the poorly aligned region is on the C-terminus of multiple aligned sequences?

Reviewer #1 (Remarks to the Author):

I feel that the authors have done a nice job of responding to my initial concerns, with one exception. They've missed the point about phosphate.

Perhaps best articulated by Tyrrell (Nature, 400, 525–531, 1999), P is considered the "ultimate" limiting nutrient, not for productivity, but for organic matter burial. The authors miss this point in their response to my comments arising from line 57 in the original manuscript when they say "Indeed, many previous studies have suggested that the increase in nutrient supply would promote primary productivity and therefore the burial of organic matter, enhancing the accumulation of O₂." The point is that P is buried with organic matter, so the rate of burial can be limited by the supply of P to the ocean (modified by the C/P ratio of buried organic matter, which I bring up later w.r.t. line 71). It's not that increasing P delivery increases primary (or even net production); indeed, these can be independent of P. So the authors are assuming that P supply does not limit organic matter burial (net oxygen production) and so should state that explicitly. Only if increasing burial efficiency increases the C/P of buried organic matter can it lead to an increase in net oxygen production.

Lee Kump

Authors' Response:

Thank you very much for your constructive comments and further clarification. We now understand that your point is that the C/P ratio of the buried organic matter and therefore the net production of O₂ can be influenced by the supply of phosphorus to the ocean.

To address this issue and explicitly state our assumption that the burial of carbon is not limited by P, we have added the following sentence in the new manuscript (Line 130 - Line 133):

“Moreover, our theory focuses on the carbon-oxygen system and assumes that the amount of organic-bound P buried with sinking organic matter remains unchanged, which implies that the C/P ratio of buried organic matter and therefore the net production of O₂ are not limited by the supply of P to the ocean.”

Reviewer #2 (Remarks to the Author):

The revised manuscript on Oxidative metabolisms catalyzed Earth's oxygenation by Shang, Rothman and Fournier was completely updated and improved according to previous comments of reviewers. However, there are still some aspects that need further clarification. In principle, they can be classified in two divisions: first the aspect of presentation of some results and second the methodological aspect.

Authors' Response:

Thank you very much for your thoughtful and constructive comments.

To solve the hard conundrum of POOM, as authors describe in the abstract on line 6 the results need to be presented more globally (to my opinion). Yes, authors made already several improvements in this respect but e.g. Figures 3 & 4 are still somehow narrowly focused (maybe this is enough for some specific journal but here a more comprehensive view is expected). Therefore, it is suggested that in Figure 3A not just the ancestor of SAR202 and its descendants are presented. There are really many HGT events demonstrated also with a high statistical support but at least it should be stated whether such HGTs are specific only for SAR202 bacteria or are general property of this gene family for (almost) all marine bacterial phyla? From current presentation it is not really obvious. Figure 4 – for a reader not deeply involved in this topic it is hard to imagine the significance of this output. At least a comparison of this diversification rate of SAR202 BVMO genes with some other (metabolically similar) gene family (presented in other literature?) is needed to get an opinion if such a diversification is comparable with other protein coding families?

Authors' Response:

Our manuscript only shows the chronogram of SAR202 cluster in Figure 3A because previous studies have only clearly shown that the SAR202 bacteria relevance to the POOM formation in marine environments [please refer to references (13), (32), (33) and (34) cited in the manuscript], which are also discussed in the second paragraph in the “Phylogenetic Analyses” section in the manuscript. The other two phyla that have BVMOs in our study – that is, Actinobacteria and Proteobacteria – have not been clearly demonstrated to play important role in POOM production, at least according to the currently available literatures (as we discussed in detail in our previous responses). But we do agree that we should clearly state that these HGTs events are not only specific to the SAR202 cluster and point out that the ancestors of Actinobacteria and Proteobacteria might be potentially important contributors to POOM production in deep time. Since this potential role of the ancestors of Actinobacteria and Proteobacteria is a speculation, we think that we should not show it as a major finding of our study (like Figure 3A). Instead, we have modified/added a few sentences in the manuscript to address this point, as you suggested in your next comment. Since we have integrated these modifications with other revisions addressing your next comment, please refer to our responses to your next comment for the modifications we have made. Moreover, Figure 3A is actually a subtree of the whole species tree chronogram (i.e., Figure S4 in the *Supplementary Information*). We have modified the following sentence in the caption of Figure 3 to inform the readers that they can find the complete species tree chronogram in the *Supplementary Information*: “*Inset A shows a subtree of calibrated chronogram showing SAR202 (blue) and related Dehalococcoidia group (red); the complete chronogram is provided in Supplementary Materials, Figure S4.*”

Regarding your suggestion about the comparison of the diversification rates of the SAR202 BVMO genes with the diversification rates of other gene family presented in the literatures, the analyses of geologic-timescale diversification rates in almost all previous studies were for animals and plants; the published work about the geologic-timescale diversification rates of microbes is very rare. A recent, comprehensive study of the diversification of microorganisms (more specifically, bacteria) is Louca *et al.*, *Nature Ecology & Evolution*, 2018(2): 1458–1467. The magnitude of diversification rates presented in this work by Louca *et al.* is ~ 0.1-1 percent per Myr [please refer to Figure 2 (d) and (g)], which is comparable to the magnitude of diversification rates shown in Figure 4 in our manuscript. But please note that this study by Louca *et al.* is about the

diversification rates of microbial species rather than a specific microbial-gene family. To the best of our knowledge, our work is one of the earliest (if not the first) studies about the geologic-timescale diversification rates of a microbial-gene family; we have not found any published studies that have presented the diversification rates of a family of (metabolically similar) microbial genes.

However, we indeed recognized the importance of testing and demonstrating the significance of the SAR202 BVMO diversification rates represented in Figure 4. Actually, this is the reason why we constructed the null model of diversification rates and performed the power spectral analysis, as discussed in the last paragraph of the “Phylogenetic Analyses” section and the “Divergence rates of BVMO genes” part of the “Materials and Methods” section in the manuscript. This null model is also an innovative part of this manuscript; we have not found any published studies performed such an analysis to test the significance of their diversification rates. As demonstrated in Figure S8 in the *Supplementary Information*, the diversification rates of the SAR202 BVMOs are very different from that of the null hypothesis (i.e., white noise). This comparison with the null model supports that the diversification rates presented in Figure 4 are meaningful biogeochemical signals rather than random noise. Moreover, we think the null model is a more powerful approach of testing the significance of SAR202 BVMO diversification than comparing them with the published results. This is not only because that the latter depends on the availability of the published work, but also because the tests using the former (i.e., the null model) are not biased by the specific genes selected for the comparisons.

And there still remains the (main) question whether SAR202 bacteria were the main (or even sole) contributors for a massive POOM burial? There are numerous cyanobacterial marine species. Could have also their ancestors (that should appear rather closely related with Chloroflexi due to several global reconstructions) contributed to POOM burial to some extent? In stromatolites they are present significantly. This does not mean that authors need to extend their phylogenetic analysis for this particular manuscript. Just few sentences about BVMO genes outside the SAR202 group would be appreciated (to follow their origin before multiple HGT events already presented here). In Figure S5 we indeed see “CMS group” that is sister clade with Chloroflexi but unfortunately in Figure S6 that shall refer to BVMO genes it is not as clearly labelled and the font used is really very small...

Authors' Response:

The manuscript focuses on the SAR202 bacteria for their demonstrated ecological/geochemical importance and relevance to the POOM formation in marine environments, as discussed in the second paragraph in the “Phylogenetic Analyses” section in the manuscript. However, we did not claim that the SAR202 bacteria are the sole contributors to the formation of POOM. Also, as we mentioned in the responses to your first comment, based on the currently available studies, no clear evidence has shown that the extant Proteobacteria and Actinobacteria species in Earth's *modern* environments are the major contributors to the production of POOM although they have BVMO genes.

More importantly, our hypothesis testing (for the POOM hypothesis) does not depend upon demonstrating that SAR202 are the lineage primarily responsible for POOM production across Earth History; rather, the POOM hypothesis predicts that known marine POOM producing lineages (i.e., SAR202) should have an evolutionary history reflecting an expanded ecological role during periods of oxygenation, and we show this prediction to be supported by the phylogenomic

data. Moreover, we cannot account for the presence of other microbial groups performing similar POOM-producing metabolisms, that may since they have gone extinct, or lost the genes for POOM production; this is true for any phylogenomic investigation of past ecologies. However, per the above argument about the “hypothesis testing” (i.e., the first half of this paragraph), this untestability does not confound the observations that can be made for surviving lineages such as the SAR202 bacteria.

Regarding your questions about cyanobacteria, indeed they are abundant in the marine environment and are very old organisms on Earth, but they are autotrophs that perform oxygenic photosynthesis and nitrogen fixation rather than heterotrophs that degrade organic matter. Moreover, the CMS (i.e., Cyanobacteria-Melainobacteria-Sericytochromatia) group and the FCB (i.e., Fibrobacteres-Chlorobi-Bacteroidete) group have not been shown to have BVMO genes and be able to catalyze the formation POOM, at least according to the currently available literatures. Therefore, the gene tree in our study does not have the BVMOs from these two groups. However, we include these two groups on the species tree for multiple purposes. First, as discussed in the manuscript (Line 329 - Line 332 in the old version; Line 343 - Line 346 in the new version), including these two groups can help us to root the species tree, because previous studies and outgroup rootings have consistently supported a species tree rooting of Bacteria where Proteobacteria, Bacteroidetes, Ignavibacteria and Chlorobi are grouped in one clade, while Actinobacteria, Chloroflexi and Cyanobacteria are grouped in the other clade. Second, the ages of the ancient cyanobacteria have been extensively studied [e.g., Magnabosco *et al.*, *Geobiology*, 2018(2): 179-189; Fournier, *et al.*, *Proceedings of the Royal Society B*, 2021(288): 20210675]; including them on the species tree help to improve the molecular-clock analyses. Third, including more closely related taxa can improve the reliability and robustness of the reconstructed phylogenies.

However, we do agree that the ancestors of some other non-SAR202 species in our phylogenies that have BVMOs – that is, Actinobacteria, and Proteobacteria – might have been more important to the formation of POOM in Earth’s ancient O₂-deficient environment compared to their modern descendants. Therefore, according to your suggestions, we have made the following changes in the new manuscript:

(1) The sentences from Line 162 to Line 163 in the old manuscript have been changed to the following sentences in the new manuscript (Line 162 - Line 164):

“Here, we reconstruct the evolutionary history of BVMOs in the SAR202 bacteria and their closely related microbial species to test the hypothesis that POOM-producing oxidative metabolisms and Earth’s oxygenation are temporally correlated.”

(2) After the paragraph about the earliest HGT acquisition, we have used a separate paragraph in the new manuscript to discuss the other HGTs between/within the Chloroflexi, Actinobacteria, and Proteobacteria phyla and have added/modified the following sentences at the beginning of this paragraph (Line 195 - Line 201):

“Fig.3 also shows that extensive HGT events of BVMO genes between/within the Chloroflexi, Actinobacteria, and Proteobacteria phyla (Supplementary Materials, Table S6) span the Proterozoic and Phanerozoic, apparently increasing in frequency starting in the Late Neoproterozoic. Although the extant taxa in the Actinobacteria, and Proteobacteria phyla have not

been demonstrated to be predominant in the production of POOM in Earth's modern environment, their ancestors might have been more important in this ecological role in the past when the O₂-limited marine environments were more extensive."

(3) We have updated the following sentence in the caption of Figure 3 in the new manuscript:

"The main figure shows a graphic summary for the data of the HGT events between/within the Chloroflexi, Actinobacteria, and Proteobacteria phyla presented in Supplementary Materials, Table S5."

(4) The following new sentence has been added into the Discussion section in the new manuscript (Line 282 - Line 285):

"Moreover, the ancestors of the Actinobacteria and Proteobacteria phyla are expected to have contributed to the formation of POOM in deep time (as discussed in the Phylogenetic Analyses section); further confirmation of this speculation by laboratory/field investigations would support the POOM hypothesis."

Regarding your concern about Figure S6, the names of phyla were not labelled in this figure because many BVMOs from the same phyla are not grouped together on this gene tree; this actually reflects that there are many HGT events on the BVMO gene tree. In the new *Supplementary Information*, we have colored the taxa names in the tips of the gene tree in Figure S6 and updated its caption to convey the information of HGTs. The other two trees of BVMO genes – that is, Figure S2 and S4 in the *Supplementary Information*, have also been updated in the same way.

Minor points about the methodology for consideration:

Lines 303-304: "Protein Basic Local Alignment Search Tool (BLASTp) on the National Center for Biotechnology Information (NCBI) database was used to search the genes of interest." But the BLASTp option is dedicated for search of PROTEINS not genes. Also the output is then formed with protein files (sequences). Have the authors searched further really for genes or just for encoded proteins? This needs to be clarified.

Authors' Response:

Thanks for pointing it out. Indeed, we used BLASTp to search for proteins rather than genes. We have corrected this in the new manuscript (Line 316 - Line 317):

"Protein Basic Local Alignment Search Tool (BLASTp) on the National Center for Biotechnology Information (NCBI) database was used to search the proteins of interest."

Lines 310-314 it is now appreciated that for those oxygenase sequences that were not well annotated in the original database homology models were produced to verify their suitability. But it is not clear and it shall be specified (at least roughly) how many from those 330 (verified) protein sequences do not belong to the SAR202 cluster?

Authors' Response:

To address your concern, we have added the following sentence in the new manuscript (Line 321 - Line 323):

"To reconstruct the gene tree of BVMOs, we used 330 protein sequences homologous to the BVMO

of SAR202 cluster bacterium Io17-Chloro-G4 (NCBI Query ID: PKB68843.1); 31 of these BVMOs belong to the SAR202 cluster.”

Lines 318-319 “poorly aligned regions were manually deleted” – this can eventually be an important aspect. If we refer to Phyre2-homology model from Figure S9 where also a pairwise alignment is present – can we at least expect that the poorly aligned region is on the C-terminus of multiple aligned sequences?

Authors’ Response:

We agree that the C-terminal regions of the proteins included high variability contributing to poor alignment and were trimmed. This is a routine step in curating alignments for phylogenetic reconstruction, designed to improve tree reconstruction by avoiding spurious site alignments. It is very frequently the case that C-terminal regions are highly divergent in protein alignments, and we have no basis for inferring that the variability of this region of the protein impacted our homology model in Fig.S9. We have specified this in the new manuscript (Line 331 - Line 333): *“Alignments were visualized on Clustal X (54), and poorly aligned regions, which primarily consist of highly variable C-terminal regions, were manually deleted.”*

Reviewers' Comments:

Reviewer #1:

Remarks to the Author:

I feel that the authors have satisfactorily addressed all my previous concerns and now advocate for publication.

Reviewer #2:

Remarks to the Author:

Authors Shang, Rothman and Fournier have improved the remaining shortcomings in their second, already improved version of the manuscript on Oxidative metabolisms catalyzed Earth's oxygenation.

It is appreciated that the legend to Figure 3 is now more informative with respect to horizontal gene transfer events and also the text on page 11 is extended in this respect.

During their further analyses, the authors have seen that indeed the diversification rates for specific microbial-gene families are rather rare in current literature. This means that presented results on specific diversification rates are a valuable contribution for hopefully frequent future comparisons.

It is also obvious, as stated on page 16 that more investigations are needed for evaluation of the potential role of some deep ancestors to the formation of POOM.

Also all minor technical details were improved on pages 17-18.

Thus, it can be now recommended to accept this manuscript in present form.

Reviewer #1 (Remarks to the Author):

I feel that the authors have satisfactorily addressed all my previous concerns and now advocate for publication.

Authors' Response:

Thank you very much for your thoughtful and constructive comments, which helped us to improve our manuscript.

Reviewer #2 (Remarks to the Author):

Authors Shang, Rothman and Fournier have improved the remaining shortcomings in their second, already improved version of the manuscript on Oxidative metabolisms catalyzed Earth's oxygenation.

It is appreciated that the legend to Figure 3 is now more informative with respect to horizontal gene transfer events and also the text on page 11 is extended in this respect.

During their further analyses, the authors have seen that indeed the diversification rates for specific microbial-gene families are rather rare in current literature. This means that presented results on specific diversification rates are a valuable contribution for hopefully frequent future comparisons. It is also obvious, as stated on page 16 that more investigations are needed for evaluation of the potential role of some deep ancestors to the formation of POOM.

Also all minor technical details were improved on pages 17-18.

Thus, it can be now recommended to accept this manuscript in present form.

Authors' Response:

Thank you very much for your thoughtful and constructive comments, which helped us to improve our manuscript.